# Improving Temporal Link Prediction via Temporal Walk Matrix Projection

**Xiaodong Lu**
CCSE Lab, Beihang University
Beijing, China
xiaodonglu@buaa.edu.cn

**Leilei Sun** *
CCSE Lab, Beihang University
Beijing, China
leileisun@buaa.edu.cn

**Tongyu Zhu**
CCSE Lab, Beihang University
Beijing, China
zhutongyu@buaa.edu.cn

**Weifeng Lv**
CCSE Lab, Beihang University
Beijing, China
lwf@buaa.edu.cn

## Abstract

Temporal link prediction, aiming at predicting future interactions among entities based on historical interactions, is crucial for a series of real-world applications. Although previous methods have demonstrated the importance of relative encodings for effective temporal link prediction, computational efficiency remains a major concern in constructing these encodings. Moreover, existing relative encodings are usually constructed based on structural connectivity, where temporal information is seldom considered. To address the aforementioned issues, we first analyze existing relative encodings and unify them as a function of temporal walk matrices. This unification establishes a connection between relative encodings and temporal walk matrices, providing a more principled way for analyzing and designing relative encodings. Based on this analysis, we propose a new temporal graph neural network called TPNet, which introduces a temporal walk matrix that incorporates the time decay effect to simultaneously consider both temporal and structural information. Moreover, TPNet designs a random feature propagation mechanism with theoretical guarantees to implicitly maintain the temporal walk matrices, which improves the computation and storage efficiency. Experimental results on 13 benchmark datasets verify the effectiveness and efficiency of TPNet, where TPNet outperforms other baselines on most datasets and achieves a maximum speedup of $33.3\times$ compared to the SOTA baseline. Our code can be found at https://github.com/lxd99/TPNet.

## 1 Intorduction

Many real-world dynamic systems can be abstracted as a temporal graph [1], where entities and interactions among them are denoted as nodes and edges with timestamps respectively. Temporal link prediction, aiming at predicting future interactions based on historical interactions, is a fundamental task for temporal graph learning, which can not only help us understand the evolution pattern of the temporal graph but also is crucial for a series of real-world tasks such as recommendations for online platforms [2, 3] and information diffusion prediction [4, 5].

Relative encodings have become an indispensable module for effective temporal link prediction [6–9] where, without them, node representations computed independently by neighbor aggregation will fail

---

*Corresponding Author.

38th Conference on Neural Information Processing Systems (NeurIPS 2024).

to capture the pairwise information. As the toy example shown in Figure 1, A and F will have the same node representation due to sharing the same local structure. Thus it can not be determined whether D will interact with A or F at $t_3$ according to their representations. However, by assigning nodes with relative encodings (i.e., additional node features) specific to the target link before computing the node representation, we can highlight the importance of each node and guide the representation learning process to extract pairwise information. For example, in Figure 1, we can infer from the relative encoding of E (in red circle) that D is more likely to interact with F than with A since D and F share a common neighbor, E (detailed discussed in Section 2.2). Although achieving remarkable success, injecting pairwise information based on relative encodings is still far from satisfactory.

1) **On Concept**, existing ways of constructing relative encodings are fragmented as they are derived from different heuristics. There still lacks a unified view on the connections between different relative encodings, which may allow a more principled way to design new relative encodings. 2) **On Method Design**, most existing relative encodings are constructed based on structural connectivity between nodes (e.g., whether two nodes are neighbors), while the temporal information is ignored. 3) **On Computation**, existing methods for relative encodings are inefficient, which usually involve time-consuming graph query operations and need to re-extract the relative encoding from scratch for each target link, making them even become the main computational bottleneck for some methods (shown in Section 4.2).

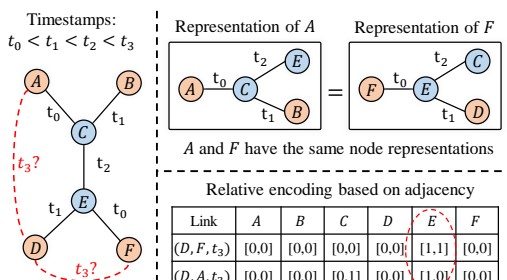

Figure 1: Without relative encodings, the learned node representations fail to capture the correlation between nodes. (For each link $(u, v, t)$, the relative encoding here for a node $w$ is $[g(u, w), g(v, w)]$, where $g(u, w) = 1$ if there is an interaction between u and w before t, otherwise $g(u, w) = 0$.)

To tackle the above issues, this paper makes the following three technical contributions. 1) **A Unified View (Concept)**. We analyze the construction of existing relative encodings and find that they can be uniformly viewed as a function of temporal random walk matrices, where different relative encodings essentially correspond to the construction of a series of temporal walk matrices based on temporal walk weighting. The presented function provides a unified view to analyze existing methods and may allow a more principal way to design new relative encodings. 2) **A Effective and Efficient Method (Method Design and Computation)**. Based on our analysis, we propose a Time decay matrix Projection-based graph neural Network for temporal link prediction, named TPNet for short. TPNet introduces a new temporal walk matrix that incorporates the time decay effect of the temporal graph, simultaneously considering both temporal and structural information. Moreover, TPNet encodes the temporal walk matrix into a series of node representations by random feature propagation, which can be efficiently updated when the graph structure changes and is storage-efficient. Importantly, such node representations can be shared by different link likelihood computation processes to decode the pairwise information, reducing the redundancy computation of re-extracting the relative encodings and avoiding time-consuming graph query operations. 3) **Theoretical and Empirical Analysis.** We conduct a theoretical analysis of the node representations obtained through random feature propagation. Our analysis demonstrates that these representations stem from the random projection of the proposed temporal walk matrix while preserving the inner product of different rows of the matrix. Moreover, we discuss the conditions under which random feature propagation can be applied to improve the computational and storage efficiency of other temporal walk matrices and provide the corresponding propagation mechanisms for temporal walk matrices of existing methods. Empirically, we conduct experiments on 13 benchmark datasets to verify the effectiveness and efficiency of the proposed method, where TPNet outperforms other baselines on most datasets and achieves a maximum speedup of $33.3\times$ compared to the SOTA baseline. Detailed ablation studies also verify the effectiveness of the proposed submodules.

## 2 Preliminary

In this section, we will first formally define some important concepts in this paper and then give a unified formulation of existing relative encodings.

## 2.1 Definitons

**Definition 1** (Temporal Graph). A temporal graph can be considered as a sequence of non-decreasing chronological interactions $\mathcal{G} = [(\{u_1, v_1\}, t_1), (\{u_2, v_2\}, t_2), \cdots]$ with $0 \leq t_1 \leq t_2 \leq \cdots$, where $(\{u_i, v_i\}, t_i)$ can be considered as a undirected link between $u_i$ and $v_i$ with timestamp $t_i$. Each node $u$ can be associated with node feature $\boldsymbol{x}_u \in \mathbb{R}^{d_N}$, and each interaction $(\{u, v\}, t)$ has link feature $\boldsymbol{e}_{u,v}^t \in \mathbb{R}^{d_E}$, where $d_N$ and $d_E$ denote the dimensions of the node feature and link feature.

**Definition 2** (Temporal Link Prediction). The interaction sequence reflects the graph dynamics, and thus the ability of a model to capture the evolution pattern of a dynamic graph can be evaluated by how accurately it predicts the future interactions based on historical interactions. Formally, given the interactions before $t$ (i.e., $\{(\{u', v'\}, t') | t' < t\}$) and two nodes $u$, $v$, the temporal link prediction task aims to predict whether there will be an interaction between $u$ and $v$ at $t$.

**Definition 3** (K-hop Subgraph). We use the notation $\mathcal{G}(t) = (\mathcal{V}(t), \mathcal{E}(t))$ to denote the graph snapshot at $t$, where $\mathcal{E}(t)$ includes all the interactions that happen before $t$ and $\mathcal{V}(t)$ includes all the nodes appear in $\mathcal{E}(t)$. Besides, we defined the k-hop subgraph of node $u$ as $\mathcal{G}_u^k(t) = (\mathcal{V}_u^k(t), \mathcal{E}_u^k(t)$, where $\mathcal{V}_u^k(t) \subset \mathcal{N}(t)$ is the set of nodes whose shortest path distance to $u$ is less than $k$ on $\mathcal{G}(t)$ and $\mathcal{E}_u^k(t) \subset \mathcal{E}(t)$ is the set of interactions between $\mathcal{V}_u^k(t)$.

**Definition 4** (Temporal Walk). A k-step temporal walk $W$ on $\mathcal{G}(t)$ is a sequence of node-time pair with decreased temporal order [6], which can be denoted as $W = [(w_0, t_0), \cdots, (w_k, t_k)]$ with $t = t_0 > t_1 > \cdots > t_k$ and $(\{w_i, w_{i+1}\}, t_{i+1})$ is an edge on $\mathcal{G}(t)$ for $0 \leq i \leq k - 1$. Figure 2 shows a visual illustration of a temporal walk. Here, we stipulate the first timestamp $t_0$ as the current time $t$ to avoid undefinedness of $t_0$. We use the notation $\mathcal{M}_{u,v}^k(t)$ to denote the set of all k-step temporal walks from $u$ to $v$ on $\mathcal{G}(t)$. Specially, $\mathcal{M}_{u,w}^0(t) = \{[(u, t)]\}$ if $u = w$ and $\mathcal{M}_{u,w}^0(t) = \emptyset$ otherwise. When there is no ambiguity, we replace $\mathcal{M}_{u,v}^k(t)$ with $\mathcal{M}_{u,v}^k$.

## 2.2 Relative Encoding for Dynamic Link Prediction

### 2.2.1 Unified Framework

Given a future link $(u, v, t)$ to be predicted, existing temporal link prediction methods (referred as node-wise methods) usually first learn the node representations $\boldsymbol{h}_u(t)$ and $\boldsymbol{h}_v(t)$ independently, which are obtained by encoding their k-hop subgraphs,

$$\boldsymbol{h}_u(t) = f_{\text{enc}}(\mathcal{G}_u^k(t), \boldsymbol{X}_{u,k}^N, \boldsymbol{X}_{u,k}^E), \quad \boldsymbol{h}_v(t) = f_{\text{enc}}(\mathcal{G}_v^k(t), \boldsymbol{X}_{v,k}^N, \boldsymbol{X}_{v,k}^E), \tag{1}$$

where $\boldsymbol{X}_{u,k}^N$ and $\boldsymbol{X}_{u,k}^E$ are the features of nodes and edges in $\mathcal{G}_u^k(t)$ respectively [2]. The $f_{\text{enc}}(\cdot)$ here can be any encoding function that maps a graph into a representation such as a k-layer GNN with a pooling layer Then the link likelihood $p_{u,v}^t$ is given $p_{u,v}^t = f_{\text{dec}}(\boldsymbol{h}_u(t), \boldsymbol{h}_v(t))$. The $f_{\text{dec}}(\cdot)$ is a decoding function that maps the node representations into link likelihood such as an MLP with a Sigmoid output layer. Detailed discussion about existing methods can be found in Appendix F.2.

As mentioned in the Section 1, learning representations independently might fail to capture the pairwise information of the given link. Thus recent methods (referred as link-wise methods) inject the pairwise information by constructing a relative encoding $\boldsymbol{r}^{w|(u,v)}$ for each node $w \in \mathcal{V}_u^k(t) \cup \mathcal{V}_v^k(t)$ as an additional node feature (Detailed construction way for different methods will be introduced in Section 2.2.2). Then Equation (1) will be changed into

$$\boldsymbol{h}_u(t) = f_{\text{enc}}(\mathcal{G}_u^k(t), \boldsymbol{X}_{u,k}^N \oplus \boldsymbol{X}_{u,k}^R, \boldsymbol{X}_{u,k}^E), \quad \boldsymbol{h}_v(t) = f_{\text{enc}}(\mathcal{G}_v^k(t), \boldsymbol{X}_{v,k}^N \oplus \boldsymbol{X}_{v,k}^R, \boldsymbol{X}_{v,k}^E), \tag{2}$$

where $\boldsymbol{X}_{u,k}^R$ is the relative encodings for nodes in $\mathcal{G}_u^k(t)$ and $\boldsymbol{X}_{u,k}^N \oplus \boldsymbol{X}_{u,k}^R$ indicate the new node features obtained by concatenating $\boldsymbol{X}_{u,k}^N$ and $\boldsymbol{X}_{u,k}^R$. Intuitively, the relative encoding $\boldsymbol{r}^{w|(u,v)}$ for each node $w$ reflects its importance to predicted link $(u, v, t)$, which can guide the representation learning process to extract the pairwise information specific to the predicted link from the subgraph. For the detailed construction way, the relative encoding $\boldsymbol{r}^{w|(u,v)}$ is the concatenation of two similarity features $\boldsymbol{r}^{w|u}$ and $\boldsymbol{r}^{w|v}$, where $\boldsymbol{r}^{w|u}/\boldsymbol{r}^{w|v}$ encodes some form of similarity between $u/v$ and $w$ (e.g., the number of k-step temporal walks from $u$ to $w$). Although the designs of similarity feature $\boldsymbol{r}^{w|u}$

---

[2]For memory-based methods (i.e, TGN), $\boldsymbol{X}_{u,k}^N$ represents the concatenation of node memories and features.

for different methods are induced from different heuristics, we find that they can be unified in a function about the temporal walk matrix, which is

$$\boldsymbol{r}^{w|u} = g([A_{u,w}^{(0)}(t), A_{u,w}^{(1)}(t), \cdots, A_{u,w}^{(k)}(t)]), \quad A_{u,w}^{(i)}(t) = \sum_{W \in \mathcal{M}_{u,w}^i} s(W) \text{ for } 0 \le i \le k. \quad (3)$$

The $s(\cdot)$ here is a score function that maps a temporal walk to a scalar and $\boldsymbol{A}^{(i)}(t)$ denotes an i-hop temporal walk matrix whose each entry $A_{u,w}^{(i)}$ is the sum of the score of all i-step temporal walks from $u$ to $w$. $g(\cdot)$ is a function applied on the vector of $[A_{u,w}^{(0)}(t), A_{u,w}^{(1)}(t), \cdots, A_{u,w}^{(k)}(t)] \in \mathbb{R}^{k+1}$ to extract similarity feature. The above equation shows that each relative encoding implies a construction of the temporal walk matrix based on weighting each temporal walk (i.e., $s(\cdot)$). Next, we will analyze existing relative encodings and show how they can be represented in the form of Equation (3).

### 2.2.2 Analysis of Existing Methods

Our analysis focuses on the four representative link-wise methods DyGFormer, PINT, NAT, and CAW, which cover all the link-wise methods in the benchmark of dynamic link prediction [9].

**DyGFormer** [9]. The $\boldsymbol{r}^{w|u}$ of DyGFormer is a one-dimensional vector representing the number of links between $w$ and $u$. For DyGFormer, we can first set the $s(\cdot)$ to be a one-const function (i.e., $s(W) = 1$ in for any $W$), which will make $A_{u,w}^{(k)}$ be the number of the k-step temporal walks from $u$ to $w$. Then, setting $g(\cdot)$ to be a function that selects the second number of a vector (i.e., $g([x_0, x_1, .., x_k]) = x_1$) will make Equation (3) generate the similarity feature of DyGFormer.

**PINT** [8]. The $\boldsymbol{r}^{w|u}$ of PINT is a $(k+1)$-dimensional vector, where $r_i^{w|u}$ is the number of $(i-1)$-step temporal walks from $u$ to $w$ for $1 \le i \le k+1$. Setting $s(\cdot)$ and $g(\cdot)$ can be set to a one-const function and an identity function respectively will make Equation (3) outputs the similarity feature of PINT.

**NAT** [7]. NAT maintains a series of fix-sized hash maps $s_u^{(0)}, ..., s_u^{(k)}$ and generates the $\boldsymbol{r}^{w|u}$ based on the hash maps. As proved in Appendix A.1, if the size of the hash maps becomes infinite, the $\boldsymbol{r}^{w|u}$ is equivalent to a $(k + 1)$-dimensional vector, where, for $1 \le i \le k + 1$, $r_i^{w|u} = 1$ if the shortest temporal walk from $u$ to $w$ is less than $i$; otherwise, $r_i^{w|u} = 0$. Let $h(\cdot)$ be a binary function where $h(x) = 1$ if $x > 0$ and $h(x) = 0$ otherwise. Setting the $s(\cdot)$ to be a one-const function and $g(\cdot)$ to a function of $g([x_0, x_1, ..., x_k]) = [h(\sum_{i=0}^0 x_i), h(\sum_{i=0}^1 x_i), ..., h(\sum_{i=0}^k x_i)]$ can make the Equation (3) generate the similarity feature of NAT.

**CAWN** [6]. The similarity feature $\boldsymbol{r}^{w|u}$ of CAWN reflects the probability of a node $w$ being visited when performing a temporal walk from $u$. Specifically, CAWN first samples a set of temporal walks beginning from $u$ based on a causal sampling strategy. Then, for each node $w$, the similarity feature $\boldsymbol{r}^{w|u}$ is extracted based on its occurrence in the sampled walks. As proved in Appendix A.2, the similarity feature $\boldsymbol{r}^{w|u}$ is the estimation of a $(k + 1)$-dimensional vector, where, for $1 \le i \le k + 1$, $r_i^{w|u}$ is the probability of visiting $w$ through a $(i-1)$-step temporal walk matrix based on the sampling strategy of CAWN, which can be represented as $r_i^{w|u} = \sum_{W \in \mathcal{M}_{u,w}^{i-1}} s'(W)$. The value $s'(W)$ for a given temporal walk $W = [(w_0, t_0), (w_1, t_1), ..., (w_k, t_k)]$ is defined as $\prod_{i=0}^{k-1} \frac{\exp(-\alpha(t_i - t_{i+1}))}{\sum_{(\{w', w\}, t') \in \mathcal{E}_{w_i, t_i}} \exp(-\alpha(t_i - t'))}$, where $\alpha$ is hyperparameter to control the sampling process, $\mathcal{E}_{w_i, t_i} = \{(\{w', w\}, t')|t' < t_i\}$ is the set of interactions attached to $w_i$ before $t_i$, and $\frac{\exp(-\alpha(t_i - t_{i+1}))}{\sum_{(\{w', w\}, t') \in \mathcal{E}_{w_i, t_i}}}$ can be considered as the probability of moving from $(w_i, t_i)$ to $(w_{i+1}, t_{i+1})$ in the sampling process. Setting $s(\cdot)$ to $s'(\cdot)$ and $g(\cdot)$ to be an identity function can make Equation (3) generate the similarity feature of CAWN.

**Conclusion**. According to the above analysis, we can conclude that (i) Equation (3) provides a unified view to consider the injection of pairwise information from the construction of temporal walk matrix, where different relative encodings (implicitly or explicitly) correspond to a kind of temporal walk matrix. (ii) Examining existing relative encodings from the unified view reveals their limitations. First, the relative encodings of existing methods (except CAWN) ignore the temporal information carried by each temporal walk, where their score function $s(\cdot)$ always yield 1 and the entries of the temporal walk matrix is just the count of the temporal walks. Next, although CAWN considers temporal information, they estimate the temporal walk matrix from the sampled temporal

---

**Algorithm 1:** Node Representation Maintaining ($\mathcal{G},\lambda,k,n,d_R$)

1   Initialize $\boldsymbol{H}^{(0)}, \boldsymbol{H}^{(1)}, ..., \boldsymbol{H}^{(k)} \in \mathbb{R}^{n \times d_R}$ as zero matrix ;

2   Fill $\boldsymbol{H}^{(0)}$ with entries independently drawn from $\mathcal{N}(0, \frac{1}{d_R})$ ;

3   **for** $(u, v, t) \in \mathcal{G}$ **do**

4     **for** $l = k$ **to** 1 **do**

5       $\boldsymbol{H}_u^{(l)} = \boldsymbol{H}_u^{(l)} + e^{\lambda t} * \boldsymbol{H}_v^{(l-1)}$ ;

6       $\boldsymbol{H}_v^{(l)} = \boldsymbol{H}_v^{(l)} + e^{\lambda t} * \boldsymbol{H}_u^{(l-1)}$ ;

7   Return $\boldsymbol{H}^{(0)}, \boldsymbol{H}^{(1)}, .., \boldsymbol{H}^{(k)}$;

---

Timestamps: $t_1 < t_2 < t_3 < t_t < t_4$

(a) Temporal Graph $\mathcal{G}(t)$

$[(B, t), (C, t_4), (A, t_3), (D, t_2), (F, t_1)]$

(b) A temporal walk on $\mathcal{G}(t)$

Figure 2: A illustration of the temporal walk.

walks, which needs time-consuming graph sampling and may introduce large estimation errors. In the next section, we will present a new temporal walk matrix to simultaneously consider the temporal and structural information and show how to efficiently maintain the proposed temporal walk matrix.

## 3 Methodology

TPNet mainly consists of two modules: Node Representation Maintaining (NRM) and Link Likelihood Computing (LLC). The NRM is responsible for encoding the pairwise information, which maintains a series of node representations. Such representations will be updated when new interaction occurs and can be used to decode the temporal walk information between two nodes. The LLC module is a prediction module, which utilizes the maintained node representations and auxiliary information (e.g., like features) to compute the likelihood of the link to be predicted.

### 3.1 Node Representation Maintaining

**Temporal Matrix Construction**. Based on the Equation (3), designing a temporal walk matrix relies on the definition of the score function $s(\cdot)$, where the element of a temporal walk matrix is $A_{u,v}^{(k)}(t) = \sum_{W \in M_{u,v}^k} s(W)$. Unlike most previous methods that only count the number of temporal walks, we consider the temporal information carried by a temporal walk in $s(\cdot)$ to simultaneously consider the temporal and structural information. Formally, let $t$ be the current time, given a temporal walk $W = [(w_0, t_0), (w_1, t_1), .., (w_k, t_k)]$, the value of the score function is $s(W) = \prod_{i=1}^{k} e^{-\lambda(t-t_i)}$, where $\lambda > 0$ is a hyperparameter to control the time decay weight. As the current time $t$ goes on, for each interaction $(\{w_i, w_{i+1}\}, t_{i+1})$ in the temporal walk $W$, its weight $e^{-\lambda(t-t_i)}$ will decay exponentially. The design of the score function is motivated by the widely observed time decay effect [1, 10] on the temporal graph, where the importance of interactions will decay as time goes on, benefiting better modeling the graph evolution patterns. In the following part of Section 3, the notation of $s(\cdot)$ and $\boldsymbol{A}^{(k)}(t)$ will specifically refer to the score function and temporal walk matrix proposed in this part. Besides, for $\boldsymbol{A}^{(0)}(t)$, we stipulate it as an identity matrix constantly.

**Node Representation Maintaining**. Directly computing the temporal walk matrices is impractical since we need to enumerate the temporal walks and each matrix needs expensive $O(n \times n)$ space complexity to store. Thus, we implicitly maintain the temporal walk matrices by maintaining a series of node representations $\boldsymbol{H}^{(0)}(t), \boldsymbol{H}^{(1)}(t), ..., \boldsymbol{H}^{(k)}(t) \in \mathbb{R}^{n \times d_R}$, where $d_R \ll n$ is the dimension of node representations and $\boldsymbol{H}_u^{(l)}(t) \in \mathbb{R}^{d_R}$ encodes the information about the $l$-step temporal walks beginning from $u$ for $0 \leq l \leq k$. The node representations will be updated when a new interaction occurs. Specifically, we first construct a random feature matrix $\boldsymbol{P} \in \mathbb{R}^{n \times d_R}$, where each entry of $\boldsymbol{P}$ is independently drawn from Gaussian distribution with mean 0 and variance $\frac{1}{d_R}$. Then we initialize $\boldsymbol{H}^{(0)}(t)$ as $\boldsymbol{P}$ and $\boldsymbol{H}^{(1)}(t), \boldsymbol{H}^{(2)}(t), ..., \boldsymbol{H}^{(k)}(t)$ as zero matrix. When a new interaction $(u, v, t)$ occurs, for $1 \leq l \leq k$, we update the representations of $u$ and $v$ by

$$\boldsymbol{H}_u^{(l)}(t^+) = \boldsymbol{H}_u^{(l)}(t) + e^{\lambda t} * \boldsymbol{H}_v^{(l-1)}(t), \quad \boldsymbol{H}_v^{(l)}(t^+) = \boldsymbol{H}_v^{(l)}(t) + e^{\lambda t} * \boldsymbol{H}_u^{(l-1)}(t), \quad (4)$$

where the $t^+$ denotes the time right after $t$. A pseudocode for maintaining the node representations is shown in Algorithm 1. The maintaining mechanism here can be considered as a random feature propagation mechanism on the temporal graph, where we initialize the zero layer's representation

of each node as a random feature and repeatedly propagate these features among nodes from the low layer to the high layer as interactions continuously appear. The following theorem shows the relationship between the node representations and the temporal walk matrices.

**Theorem 1.** *If any two interactions on temporal graph $\mathcal{G}$ have different timestamps, the obtained representations $\boldsymbol{H}^{(0)}(t), \boldsymbol{H}^{(1)}(t), ..., \boldsymbol{H}^{(k)}(t)$ satisfy $e^{-\lambda l t} * \boldsymbol{H}^{(l)}(t) = \boldsymbol{A}^{(l)}(t)\boldsymbol{P}$ for $0 \le l \le k$.*

For simplicity, we assume the timestamps of the interaction are different, and we show how to update the representations when multiple interactions have the same timestamps in Appendix C. We leave the proof in the Appendix B.1. The above theorem shows that the obtained node representation is the projection (i.e. linear transformation) of the temporal walk matrices, where the transform matrix is the initial random feature matrix $\boldsymbol{P}$. The next theorem shows that the node representations preserve the inner product of the temporal walk matrices.

**Theorem 2.** *Given any $\epsilon \in (0,1)$, let $\| \cdot \|_2$ denote the L2 norm, $c_{u,v}^{l_1,l_2}$ denote $\frac{1}{2}(\|\boldsymbol{A}_u^{(l_1)}(t)\|_2^2 + \|\boldsymbol{A}_v^{(l_2)}(t)\|_2^2)$, and $\bar{\boldsymbol{H}}^{(l)}(t)$ denote $e^{-\lambda l t} * \boldsymbol{H}^{(l)}(t)$, if dimension $d_R$ of node representations satisfy $d_R \ge \frac{24}{\epsilon^2} \log(4^{1/3}(k+1)n)$, then for any $1 \le u, v \le n$, and $0 \le l_1, l_2 \le k$, we have*

$$\mathbb{P}\left(\left|\langle\bar{\boldsymbol{H}}_u^{(l_1)}(t), \bar{\boldsymbol{H}}_v^{(l_2)}(t)\rangle - \langle\boldsymbol{A}_u^{(l_1)}(t), \boldsymbol{A}_v^{(l_2)}(t)\rangle\right| \le \epsilon c_{u,v}^{l_1,l_2}\right) \ge 1 - \frac{1}{(k+1)n}, \quad (5)$$

*where $\langle\cdot,\cdot\rangle$ denotes the inner product and $|\cdot|$ denotes taking absolute value.*

We leave the proof in Appendix B.2. The above theorem shows that the inner product of the node representations is approximately equal to the inner product of the temporal walk matrices (i.e., $\langle\bar{\boldsymbol{H}}_u^{(l_1)}(t), \bar{\boldsymbol{H}}_v^{(l_2)}(t)\rangle \approx \langle\boldsymbol{A}_u^{(l_1)}(t)), \boldsymbol{A}_v^{(l_2)}(t)\rangle$). Specifically, given any error rate $\epsilon$, if the dimension of the node representations satisfies a certain condition, the difference between $\langle\bar{\boldsymbol{H}}_u^{(l_1)}(t), \bar{\boldsymbol{H}}_v^{(l_2)}(t)\rangle$ and $\langle\boldsymbol{A}_u^{(l_1)}(t)), \boldsymbol{A}_v^{(l_2)}(t)\rangle$ for any $u,v,l_1,l_2$ will be less than $\epsilon c_{u,v}^{l_1,l_2}$ with high probability ($\ge 1 - \frac{1}{(k+1)n}$). The error rate can approach 0 infinitely and thus the inner product of the node representations can approach that of temporal walk matrices infinitely, at the cost of increasing the dimension $d_R$. As we will see in Section 4.3, a small dimension ($\ll n$) in practice is enabled to make the inner product of node representations a good estimation of that of temporal walk matrices. Additionally, due to the i-th row of $\boldsymbol{A}^{(0)}(t)$ being the one-hot encoding of i (since it is an identity matrix), we have $\langle\boldsymbol{A}_u^{(l)}(t), \boldsymbol{A}_w^{(0)}(t)\rangle = A_{u,w}^{(l)}(t)$. Thus, we can obtain $[A_{u,w}^{(0)}(t), \cdots, A_{u,w}^{(k)}(t)]$ in Equation (3) by calculating the inner product between all layers of $u$'s representation and the zero layer of $w$'s representation, expressed as $[\langle\bar{\boldsymbol{H}}_u^{(0)}(t), \bar{\boldsymbol{H}}_w^{(0)}(t)\rangle, \cdots, \langle\bar{\boldsymbol{H}}_u^{(k)}(t), \bar{\boldsymbol{H}}_w^{(0)}(t)\rangle]$.

**Remark.** Compared to directly computing the temporal walk matrices $\boldsymbol{A}^{(0)}(t), .., \boldsymbol{A}^{(k)}(t)$, which needs to enumerate the temporal walks and $O((k+1)n^2)$ space complexity to store, maintaining the node representations largely improve the computation and storage efficiency, which only needs $O((k+1)nd_R)$ space complexity to store and $O(kd_R)$ time complexity to update when a new interaction occurs. Actually, Theorem 2 is the direct result of Theorem 1 based on Johnson-Lindenstrauss Lemma [11], where the random projection can preserve the inner product and norm [12]. Notably, the method for implicitly maintaining temporal walk matrices via random feature propagation can be extended to other types of temporal walk matrices, provided they meet a specific condition (i.e., the updating function of the temporal walk matrix can be written as the linear combination of its rows). This condition is not restrictive, and all the temporal walk matrices discussed in Section 2 fulfill it. We show their propagation mechanism and related discussion in Appendix D. In conclusion, the unified function in Equation (3), combined with methods of implicitly maintaining the temporal walk matrices, provides a new way to design a more effective and efficient way to inject pairwise information.

## 3.2 Link Likelihood Computing

Given an interaction $(u,v,t)$ to be predicted, we compute its happening likelihood based on the obtained node representations and auxiliary features. Specifically, we first decode a pairwise feature $\boldsymbol{f}_{u,v}(t)$ from the node representations obtained in Section 3.1. Then we compute the node embeddings $\boldsymbol{h}_u(t)$ and $\boldsymbol{h}_v(t)$ for node $u$ and $v$ respectively based on their historical interactions. Finally, we give the link likelihood based on $\boldsymbol{h}_u(t), \boldsymbol{h}_v(t), \boldsymbol{f}_{u,v}(t)$. For notation simplicity, we omit the suffix of $\boldsymbol{h}_u(t), \boldsymbol{h}_v(t), \boldsymbol{f}_{u,v}(t)$ and denote them as $\boldsymbol{h}_u, \boldsymbol{h}_v, \boldsymbol{f}_{u,v}$ in the following part.

**Pairwise Feature Decoding**. Although we can obtain the $(k+1)$-dimensional feature in Equation (3) by calculating the inner product between the zero-layer representation of one node and all layers of the other node's representation, this method does not consider the correlation between all layers of both nodes. Therefore, we use representations from all layers to decode the pairwise information. Specifically, we first take the node representation of u and v from different layers, which can be denoted as $\boldsymbol{F}_* = [e^{-\lambda 0t}\boldsymbol{H}_*^{(0)}, ..., e^{-\lambda kt}\boldsymbol{H}_*^{(k)}] \in \mathbb{R}^{(k+1)\times d_R}$ with $*$ could be u or v. Then we concatenate them together to get $\boldsymbol{F}_{u,v} = [\boldsymbol{F}_u, \boldsymbol{F}_v] \in \mathbb{R}^{2(k+1)\times d_R}$ and obtain the raw pairwise feature $\tilde{\boldsymbol{f}}_{u,v}$ by computing the inner product among different rows of $\boldsymbol{F}_{u,v}$, which is $\tilde{\boldsymbol{f}}_{u,v} = \text{flat}(\boldsymbol{F}_{u,v}\boldsymbol{F}_{u,v}^T)$ with flat$(\cdot)$ means flatten a matrix of $\mathbb{R}^{2(k+1)\times 2(k+1)}$ into a vector of $\mathbb{R}^{4(k+1)^2}$. Finally, we feed the raw pairwise feature $\tilde{\boldsymbol{f}}_{u,v}$ into an MLP to get the pairwise feature $\boldsymbol{f}_{u,v}$, which is $\boldsymbol{f}_{u,v} = \text{MLP}(\log(\text{ReLU}(\tilde{\boldsymbol{f}}_{u,v}) + 1))$. The ReLU$(\cdot)$ here is used to reduce estimation error, where the inner product of temporal walk matrices should be larger than zero and we thus set it to be zero if the inner product of the node representations is negative. The $\log(\cdot)$ is used to scale the raw pairwise feature, where the range of the inner product between different layers varies greatly and the $+1$ is the shift term to avoid the undefined value of $\log(0)$. We will see in Section 4.3 that these two operations can improve the training stability.

**Auxiliary Feature Learning**. The auxiliary features such as link features also provide rich information for modeling the evolution patterns of the temporal graph. In this part, we follow the previous methods [13, 9] and consider the auxiliary feature learning as a sequential learning problem. Specifically, for node $u$, we take its recent $m$ interactions $S_u = [(\{u, w_1\}, t_1), ..., (\{u, w_m\}, t_m)]$ before $t$ and learn node embedding $\boldsymbol{h}_u$ from this sequence. We first fetch the node features $\boldsymbol{X}_{u,N} = [\boldsymbol{x}_{w_1}, ..., \boldsymbol{x}_{w_m}] \in \mathbb{R}^{m\times d_N}$ and edge features $\boldsymbol{X}_{u,E} = [\boldsymbol{e}_{u,w_1}^{t_1}, .., \boldsymbol{e}_{u,w_1}^{t_m}] \in \mathbb{R}^{m\times d_E}$. For timestamps, we map the timestamps into temporal features $\boldsymbol{X}_{u,T} = [\phi(t-t_1), .., \phi(t-t_n)] \in \mathbb{R}^{m\times d_T}$ like [14], where $\phi(\Delta t) = [\cos(w_1\Delta t), .., \cos(w_{d_T}\Delta t)]$ is a time encoding function to learn the periodic temporal pattern. Besides, we construct a relative encoding sequence $\boldsymbol{X}_{u,F} = [\boldsymbol{f}_{u,w_1} \oplus \boldsymbol{f}_{v,w_1}, ..., \boldsymbol{f}_{u,w_m} \oplus \boldsymbol{f}_{v,w_m}] \in \mathbb{R}^{m\times 8(k+1)^2}$ to inject the pairwise features, where $\boldsymbol{f}_{u,w_m}$ denote the pairwise feature of $u$ and $w_m$ and $\oplus$ is the concatenation operation. After obtaining the above feature sequence, we concatenate them together and feed it into an MLP to get the final feature sequence $\boldsymbol{Z}_u^{(0)} = \text{MLP}([\boldsymbol{X}_{u,N}, \boldsymbol{X}_{u,E}, \boldsymbol{X}_{u,T}, \boldsymbol{X}_{u,F}]) \in \mathbb{R}^{m\times d}$. Subsequently, we stack $l$ layers of MLP-Mixer [15] to capture the temporal and structural dependencies within the feature sequence, which is

$$
\begin{aligned}
\tilde{\boldsymbol{Z}}_u^{(l)} &= \boldsymbol{Z}_u^{(l-1)} + \boldsymbol{W}_1^{(l)}\text{GeLU}(\boldsymbol{W}_2^{(l)}\text{LayerNorm}(\boldsymbol{Z}_u^{(l-1)})) \\
\boldsymbol{Z}_u^{(l)} &= \tilde{\boldsymbol{Z}}_u^{(l)} + \boldsymbol{W}_3^{(l)}\text{GeLU}(\boldsymbol{W}_4^{(l)}\text{LayerNorm}(\tilde{\boldsymbol{Z}}_u^{(l)})).
\end{aligned}
\tag{6}
$$

Finally, we get the node embedding by mean pooling $\boldsymbol{h}_u = \text{MEAN}(\boldsymbol{Z}_u^{(l)})$. The procedure to get node embedding $\boldsymbol{h}_v$ is similar and for the node that does not have $m$ interactions, we pad the feature sequence with zero. Then, the likelihood of the link $(u, v, t)$ is given by $p_{u,v}^t = \text{MLP}([\boldsymbol{h}_u, \boldsymbol{h}_v, \boldsymbol{f}_{u,v}])$, where MLP$(\cdot)$ is a 2-layer MLP model with Sigmoid activation function in its output layer.

# 4 Experiments

## 4.1 Experimental Settings

**Datasets and Baselines**. We conduct experiments on 13 benchmark datasets for temporal link prediction, which are Wikipedia, Reddit, MOOC, LastFM, Enron, Social Evo., UCI, Flights, Can. Parl., US Legis., UN Trade, UN Vote and Contact. Eleven popular temporal graph learning methods are selected as baselines including JODIE, DyRep, TGAT, TGN, CAWN, EdgeBank, TCL, GraphMixer, NAT, PINT, and DyGFormer. Details about datasets and baselines can be found in Appendix F.

**Task Settings**. The task settings strictly follow [9]. Specifically, we conduct experiments under two settings: 1) the transductive setting that predicts links between nodes that have been seen during training and 2) the inductive setting that predicts links between nodes that are not seen during training. Three different negative sampling strategies introduced by [16] are used to sample the negative links and the Average Precision (AP) and Area Under the Receiver Operating Characteristic Curve (AUC-ROC) are adopted as the evaluation metrics. For dataset splitting, we chronologically split

Table 1: Transductive results for different baselines under the random negative sampling strategy. **blod** and underline highlight the best and second best result respectively.

| Metric | Dataset | JODIE | DyRep | TGAT | TGN | CAWN | EdgeBank | TCL | GraphMixer | NAT | PINT | DyGFormer | TPNet |
|---|---|---|---|---|---|---|---|---|---|---|---|---|---|
| **AP** | Wikipedia | $96.50_{\pm0.14}$ | $94.86_{\pm0.06}$ | $96.94_{\pm0.06}$ | $98.45_{\pm0.06}$ | $98.76_{\pm0.03}$ | $90.37_{\pm0.00}$ | $96.47_{\pm0.16}$ | $97.25_{\pm0.03}$ | $98.03_{\pm0.07}$ | $98.45_{\pm0.04}$ | $99.03_{\pm0.02}$ | **$99.32_{\pm0.03}$** |
| | Reddit | $98.31_{\pm0.14}$ | $98.22_{\pm0.04}$ | $98.52_{\pm0.02}$ | $98.63_{\pm0.06}$ | $99.11_{\pm0.01}$ | $94.86_{\pm0.00}$ | $97.53_{\pm0.02}$ | $97.31_{\pm0.01}$ | $99.13_{\pm0.10}$ | $99.15_{\pm0.02}$ | $99.22_{\pm0.01}$ | **$99.27_{\pm0.00}$** |
| | MOOC | $80.23_{\pm2.44}$ | $81.97_{\pm0.49}$ | $85.84_{\pm0.15}$ | $89.15_{\pm1.60}$ | $80.15_{\pm0.25}$ | $57.97_{\pm0.00}$ | $82.38_{\pm0.24}$ | $82.78_{\pm0.15}$ | $85.88_{\pm0.55}$ | $88.08_{\pm0.86}$ | $87.52_{\pm0.49}$ | **$96.39_{\pm0.09}$** |
| | LastFM | $70.85_{\pm2.13}$ | $71.92_{\pm2.21}$ | $73.42_{\pm0.21}$ | $77.07_{\pm3.97}$ | $86.99_{\pm0.06}$ | $79.29_{\pm0.00}$ | $67.27_{\pm2.16}$ | $75.61_{\pm0.24}$ | $88.02_{\pm1.94}$ | $89.66_{\pm1.81}$ | $93.00_{\pm0.12}$ | **$94.50_{\pm0.08}$** |
| | Enron | $84.77_{\pm0.30}$ | $82.38_{\pm3.36}$ | $71.12_{\pm0.97}$ | $86.53_{\pm1.11}$ | $89.56_{\pm0.09}$ | $83.53_{\pm0.00}$ | $79.70_{\pm0.71}$ | $82.25_{\pm0.16}$ | $90.60_{\pm0.66}$ | $92.20_{\pm0.15}$ | $92.47_{\pm0.12}$ | **$92.90_{\pm0.17}$** |
| | Social Evo. | $89.89_{\pm0.55}$ | $88.87_{\pm0.30}$ | $93.16_{\pm0.17}$ | $93.57_{\pm0.17}$ | $84.96_{\pm0.09}$ | $74.95_{\pm0.00}$ | $93.13_{\pm0.16}$ | $93.37_{\pm0.07}$ | $88.92_{\pm3.45}$ | $94.42_{\pm0.03}$ | **$94.73_{\pm0.01}$** | **$94.73_{\pm0.02}$** |
| | UCI | $89.43_{\pm1.09}$ | $65.14_{\pm2.30}$ | $79.63_{\pm0.70}$ | $92.34_{\pm1.04}$ | $95.18_{\pm0.06}$ | $76.20_{\pm0.00}$ | $89.57_{\pm1.63}$ | $93.25_{\pm0.57}$ | $93.40_{\pm0.26}$ | $96.45_{\pm0.11}$ | $95.79_{\pm0.17}$ | **$97.35_{\pm0.04}$** |
| | Flights | $95.60_{\pm1.73}$ | $95.29_{\pm0.72}$ | $94.03_{\pm0.18}$ | $97.95_{\pm0.14}$ | $98.51_{\pm0.01}$ | $89.35_{\pm0.00}$ | $91.23_{\pm0.02}$ | $90.99_{\pm0.05}$ | $98.57_{\pm0.12}$ | $98.80_{\pm0.02}$ | $98.91_{\pm0.01}$ | **$98.93_{\pm0.02}$** |
| | Can. Parl. | $69.26_{\pm0.31}$ | $66.54_{\pm2.76}$ | $70.73_{\pm0.72}$ | $70.88_{\pm2.34}$ | $69.82_{\pm2.34}$ | $64.55_{\pm0.00}$ | $68.67_{\pm2.67}$ | $77.04_{\pm0.46}$ | $79.72_{\pm1.76}$ | $68.36_{\pm1.43}$ | **$97.36_{\pm0.45}$** | $90.28_{\pm0.37}$ |
| | US Legis. | $75.05_{\pm1.52}$ | $75.34_{\pm0.39}$ | $68.52_{\pm3.16}$ | $75.99_{\pm0.58}$ | $70.58_{\pm0.48}$ | $58.39_{\pm0.00}$ | $69.59_{\pm0.48}$ | $70.74_{\pm1.02}$ | $78.71_{\pm0.87}$ | $74.85_{\pm0.97}$ | $71.11_{\pm0.59}$ | **$80.58_{\pm0.23}$** |
| | UN Trade | $64.94_{\pm0.31}$ | $63.21_{\pm0.93}$ | $61.47_{\pm0.18}$ | $65.03_{\pm1.37}$ | $65.39_{\pm0.12}$ | $60.41_{\pm0.00}$ | $62.21_{\pm0.03}$ | $62.61_{\pm0.27}$ | $73.95_{\pm1.16}$ | $70.20_{\pm0.58}$ | $66.46_{\pm1.29}$ | **$87.24_{\pm0.65}$** |
| | UN Vote | $63.91_{\pm0.81}$ | $62.81_{\pm0.80}$ | $52.21_{\pm0.98}$ | $65.72_{\pm2.17}$ | $52.84_{\pm0.10}$ | $58.49_{\pm0.00}$ | $51.90_{\pm0.30}$ | $52.11_{\pm0.16}$ | $70.45_{\pm0.68}$ | $66.25_{\pm0.78}$ | $55.55_{\pm0.42}$ | **$75.12_{\pm0.29}$** |
| | Contact | $95.31_{\pm1.33}$ | $95.98_{\pm0.15}$ | $96.28_{\pm0.09}$ | $96.89_{\pm0.56}$ | $90.26_{\pm0.28}$ | $92.58_{\pm0.00}$ | $92.44_{\pm0.12}$ | $91.92_{\pm0.03}$ | $97.39_{\pm0.22}$ | $98.64_{\pm0.02}$ | $98.29_{\pm0.01}$ | **$98.66_{\pm0.01}$** |
| | Avg. Rank | 7.85 | 8.77 | 8.54 | 5.00 | 7.00 | 10.54 | 9.77 | 8.31 | 4.15 | 3.69 | 3.15 | **1.08** |
| **AUC** | Wikipedia | $96.33_{\pm0.07}$ | $94.37_{\pm0.09}$ | $96.67_{\pm0.07}$ | $98.37_{\pm0.07}$ | $98.54_{\pm0.04}$ | $90.78_{\pm0.00}$ | $95.84_{\pm0.18}$ | $96.92_{\pm0.03}$ | $97.75_{\pm0.11}$ | $98.16_{\pm0.06}$ | $98.91_{\pm0.02}$ | **$99.30_{\pm0.02}$** |
| | Reddit | $98.31_{\pm0.05}$ | $98.17_{\pm0.05}$ | $98.47_{\pm0.02}$ | $98.60_{\pm0.06}$ | $99.01_{\pm0.01}$ | $95.37_{\pm0.00}$ | $97.42_{\pm0.02}$ | $97.17_{\pm0.02}$ | $99.09_{\pm0.10}$ | $99.09_{\pm0.03}$ | $99.15_{\pm0.01}$ | **$99.22_{\pm0.00}$** |
| | MOOC | $83.81_{\pm2.09}$ | $85.03_{\pm0.58}$ | $87.11_{\pm0.19}$ | $91.21_{\pm1.15}$ | $80.38_{\pm0.26}$ | $60.86_{\pm0.00}$ | $83.12_{\pm0.18}$ | $84.01_{\pm0.17}$ | $87.42_{\pm0.58}$ | $90.55_{\pm0.43}$ | $87.91_{\pm0.58}$ | **$97.17_{\pm0.08}$** |
| | LastFM | $70.49_{\pm1.66}$ | $71.16_{\pm1.89}$ | $71.59_{\pm0.18}$ | $78.47_{\pm2.94}$ | $85.92_{\pm0.10}$ | $83.77_{\pm0.00}$ | $73.53_{\pm0.12}$ | | $86.92_{\pm2.72}$ | $89.28_{\pm1.63}$ | $93.05_{\pm0.10}$ | **$94.39_{\pm0.04}$** |
| | Enron | $87.96_{\pm0.52}$ | $84.89_{\pm3.00}$ | $68.89_{\pm1.10}$ | $88.32_{\pm0.99}$ | $90.45_{\pm0.14}$ | $87.05_{\pm0.00}$ | $75.74_{\pm0.72}$ | $84.38_{\pm0.21}$ | $91.68_{\pm0.83}$ | $92.87_{\pm0.34}$ | $93.33_{\pm0.13}$ | **$93.98_{\pm0.26}$** |
| | Social Evo. | $92.05_{\pm0.46}$ | $90.76_{\pm0.21}$ | $94.76_{\pm0.16}$ | $95.39_{\pm0.17}$ | $87.34_{\pm0.08}$ | $81.60_{\pm0.00}$ | $94.84_{\pm0.17}$ | $95.23_{\pm0.07}$ | $90.84_{\pm3.72}$ | $96.16_{\pm0.02}$ | $96.30_{\pm0.01}$ | **$96.43_{\pm0.02}$** |
| | UCI | $90.44_{\pm0.49}$ | $68.77_{\pm2.34}$ | $78.53_{\pm0.74}$ | $92.03_{\pm1.13}$ | $93.87_{\pm0.08}$ | $77.30_{\pm0.00}$ | $87.82_{\pm1.36}$ | $91.81_{\pm0.67}$ | $92.31_{\pm0.37}$ | $95.57_{\pm0.16}$ | $94.49_{\pm0.26}$ | **$96.79_{\pm0.04}$** |
| | Flights | $96.21_{\pm1.42}$ | $95.95_{\pm0.62}$ | $94.13_{\pm0.17}$ | $98.22_{\pm0.13}$ | $98.45_{\pm0.01}$ | $90.23_{\pm0.00}$ | $91.21_{\pm0.02}$ | $91.13_{\pm0.01}$ | $98.69_{\pm0.10}$ | $98.89_{\pm0.02}$ | $98.93_{\pm0.01}$ | **$99.00_{\pm0.02}$** |
| | Can. Parl. | $78.21_{\pm0.23}$ | $73.35_{\pm3.67}$ | $75.69_{\pm0.78}$ | $76.99_{\pm1.80}$ | $75.70_{\pm3.27}$ | $64.14_{\pm0.00}$ | $72.46_{\pm3.23}$ | $83.17_{\pm0.53}$ | $84.04_{\pm1.13}$ | $77.96_{\pm1.46}$ | **$97.76_{\pm0.41}$** | $92.05_{\pm0.34}$ |
| | US Legis. | $82.85_{\pm1.07}$ | $82.28_{\pm0.32}$ | $75.84_{\pm1.99}$ | $83.34_{\pm0.43}$ | $77.16_{\pm0.39}$ | $62.57_{\pm0.00}$ | $76.27_{\pm0.63}$ | $76.96_{\pm0.79}$ | $85.36_{\pm0.52}$ | $82.10_{\pm0.85}$ | $77.90_{\pm0.58}$ | **$86.49_{\pm0.18}$** |
| | UN Trade | $69.62_{\pm0.44}$ | $67.44_{\pm0.63}$ | $64.01_{\pm0.12}$ | $69.10_{\pm1.67}$ | $68.54_{\pm0.16}$ | $64.72_{\pm0.05}$ | $65.52_{\pm0.51}$ | | $77.61_{\pm1.36}$ | $74.87_{\pm0.53}$ | $70.20_{\pm1.44}$ | **$89.17_{\pm0.46}$** |
| | UN Vote | $68.53_{\pm0.95}$ | $67.18_{\pm1.04}$ | $52.83_{\pm1.12}$ | $69.71_{\pm2.65}$ | $53.09_{\pm0.22}$ | $62.97_{\pm0.00}$ | $51.88_{\pm0.36}$ | $52.46_{\pm0.27}$ | $75.32_{\pm0.63}$ | $70.69_{\pm1.02}$ | $57.12_{\pm0.62}$ | **$79.88_{\pm0.30}$** |
| | Contact | $96.66_{\pm0.89}$ | $96.48_{\pm0.14}$ | $96.95_{\pm0.08}$ | $97.54_{\pm0.35}$ | $89.99_{\pm0.34}$ | $94.34_{\pm0.00}$ | $94.15_{\pm0.09}$ | $93.94_{\pm0.02}$ | $97.79_{\pm0.16}$ | $98.90_{\pm0.02}$ | $98.53_{\pm0.01}$ | **$98.91_{\pm0.01}$** |
| | Avg. Rank | 7.15 | 8.69 | 8.92 | 5.08 | 7.15 | 10.31 | 10.15 | 8.62 | 4.08 | 3.46 | 3.23 | **1.08** |

each dataset with $70\%/15\%/15\%$ for training/validating/testing. The training and testing pipeline is the same as that in [9].

**Model Configuration**. For TPNet, the layer $l$ of node representations, the number of recent interactions $m$, and dimension $d_R$ of the node representations are set to 3, 20 and $10 * \log(2E)$, where E is the number of the interactions. We find the best time decay weight $\lambda$ via grid search within a range of $10^{-4}$ to $10^{-7}$. Specifically, the best $\lambda$ for Contact is $10^{-4}$, the best $\lambda$ for MOOC, Can. Parl. and UN Vote is $10^{-5}$, the best $\lambda$ for Wikipedia, Reddit, Enron, Social Evo., Flights and US Legis. is $10^{-6}$, the best $\lambda$ for LastFM, UCI and UN Trade is $10^{-7}$.

**Implementatoin Details**. For baselines, we use the implementation of DyGLib [9], which is a unified temporal graph learning library, and has tuned the best hyperparameters for each baseline. For baselines that are not included in DyGLib (i.e., NAT and PINT), we use their official implementation and find the best hyperparameters via grid search. Experiments are conducted on a Ubuntu server, whose CPU and GPU devices are one Intel(R) Xeon(R) Gold 6226R CPU @ 2.9GHz with 64 CPU cores and four GeForce RTX 3090 GPUs with 24 GB memory respectively. We run each experiment five times and report the average.

## 4.2 Performance and Efficiency Comparison

**Performance comparison among baselines**. The performance of TPNet and baselines is shown in Table 1. Due to space limit, Table 1 only shows the results under the transductive setting with random negative sampling strategy and more similar results can be found in Appendix G.1. As shown in Table 1, TPNet achieves the best performance among all the methods on most datasets, verifying its effectiveness. Besides, the link-wise methods (CAWN, NAT, PINT, and DyGFormer) perform better than the node-wise methods, indicating the importance of injecting the pairwise feature. Compared to the baselines, TPNet simultaneously considers the temporal and structural correlations between nodes and encodes the temporal walk matrix into node representations with theoretical guarantees, which contributes to its superior performance.

**Efficiency Analysis**. We compare the relative inference time of different methods to TPNet to evaluate their efficiency. The results on LastFM and MOOC are shown in Figure 4 and more results can be found in Appendix G.2. As shown in Figure 4, TPNet not only achieves the best performance but is also more efficient than other link-wise methods, where TPNet achieves a $33\times$ and $71.5\times$

Table 2: Ablation study results, where N/A indicates the numerical overflow error.

| Datasets | TPNet | w/o NR | w/o Time | w/o Scale |
|---|---|---|---|---|
| MOOC | $96.39_{\pm0.09}$ | $83.21_{\pm0.23}$ | $94.62_{\pm0.34}$ | $63.04_{\pm0.95}$ |
| LastFM | $94.50_{\pm0.08}$ | $76.52_{\pm0.41}$ | $94.30_{\pm0.03}$ | N/A |
| Enron | $92.90_{\pm0.17}$ | $83.23_{\pm0.13}$ | $92.85_{\pm0.17}$ | N/A |
| UCI | $97.35_{\pm0.04}$ | $88.70_{\pm2.53}$ | $97.19_{\pm0.10}$ | $73.13_{\pm2.57}$ |
| US Legis. | $80.58_{\pm0.23}$ | $69.47_{\pm1.33}$ | $71.83_{\pm0.52}$ | $70.44_{\pm1.97}$ |
| UN Trade | $87.24_{\pm0.65}$ | $62.56_{\pm0.32}$ | $65.98_{\pm0.45}$ | $56.58_{\pm1.08}$ |
| UN Vote | $75.12_{\pm0.29}$ | $52.58_{\pm0.66}$ | $54.80_{\pm0.28}$ | $53.20_{\pm1.39}$ |

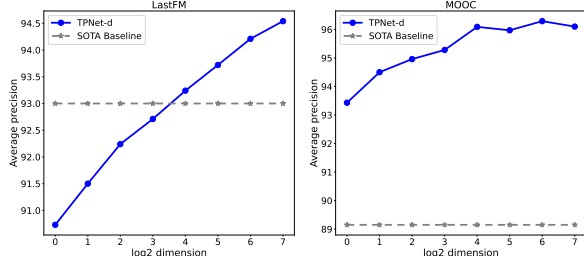

Figure 3: Influence of node representation dimension.

speedups compared to the SOTA baseline DyGFormer and CAWN respectively on LastFM. The CAWN and DyGFormer models utilize time-consuming graph queries (e.g., temporal walk sampling) to construct relative encodings, which constitute the main computational bottleneck and consume over 70% of the running time. In contrast, TPNet caches historical interactions into node representations and constructs pairwise features based on these representations, thereby enhancing efficiency.

**Scalability Analysis**. To further verify the scalability of TPNet, we generated a series of random graphs with an average degree fixed at 100 and the number of edges varying from 1e5 to 1e8. Figure 4 shows the change of running time and GPU memory of TPNet, where the growth of the running time and GPU memory is close to and less than the linear growth curve respectively, showing the good scalability of TPNet. In contrast, PINT explicitly stores the temporal walk matrices and encounters out-of-memory error when the edge number reaches 1e7 (shown inAppendix G.3), verifying the impracticability of explicitly storing temporal walk matrices.

## 4.3 Ablation Study

**Proposed Components**. To verify the effectiveness of the proposed components in TPNet, we compare TPNet with the following variants: 1) w/o NR that remove the node representations and the corresponding features that decoded from them. 2) w/o Time that only considers the structural information by setting the time decay weight $\lambda$ to be zero. 3) w/o Scale that remove the $\log(\cdot)$ and $\text{ReLU}(\cdot)$ in the pairwise feature decoding. As shown in Table 2, there is a dramatic performance drop of w/o NR, which shows that the pairwise information carried by the node representations plays a vital role in the performance of TPNet. There is also an obvious performance drop of w/o Time, which confirms the necessity of incorporating temporal information in temporal walk matrix construction. Besides, the unreasonable poor performance of the w/o Scale is due to the various distribution of node representations across different layers, where, without scaling the raw pairwise features, the training will be unstable, and numerical overflow errors may even occur on some datasets. Further details on the distribution of node representations from different layers are provided in Appendix G.5.

**Dimension Change**. To verify the influence of the node representation dimension. We vary the dimensions from 1 to 128 and report the performance of TPNet (denoted as TPNet-d). As shown in Figure 3, the required dimension of node representations is small, where only 1-dimensional and 16-dimensional node representations can achieve satisfactory performance on MOOC and LastFM respectively. For different datasets, we observe that the average degree may be a main influence of the required dimension, where sparse graphs (like MOOC and Wikipedia) only need a small dimension, and dense graphs (like LastFM and Enron) may require a larger dimension. Empirically, setting the dimension to be $10 * \log(2E)$ is enough to achieve satisfactory performance on all datasets, where $E$ is the number of edges. Results on more datasets can be found in Appendix G.4.

## 5 Related Work

Temporal link prediction [17] aims at predicting future interactions based on historical topology, which is crucial for a series of real-world applications [2, 3, 18]. Earlier methods considered the temporal graph as a series of graph snapshots that are sampled at regularly-spaced timestamps [19, 20], which will lose the fine-grained temporal information due to ignoring the temporal orders of

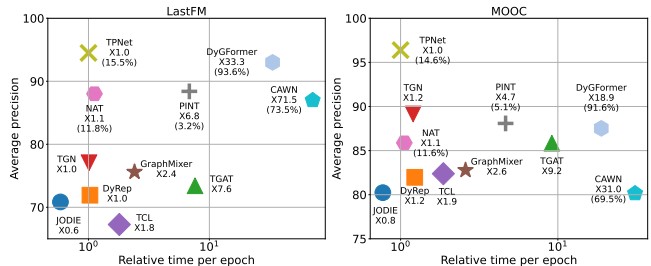
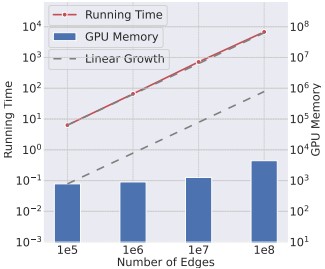

Figure 4: Relative running time of different methods. The proportion of construction relative encoding time to all running time is marked in brackets for link-wise methods.

Figure 5: Scalability analysis of TPNet on synthetic datasets with different sizes.

interactions in a graph snapshot. Recently, some continuous-time temporal graph learning methods have been proposed [21, 3, 14, 13], which consider the temporal graph as a sequence of interactions with irregular time intervals to fully capture the graph dynamics. For example, TGN [21] maintained a dynamic memory vector for each node and generated node representations by aggregating memory vectors via temporal graph attention to capture the evolution pattern of the temporal graph. However, capturing pairwise information by merely aggregating representations of neighboring nodes [22] is challenging. To address this issue, the link-wise method was proposed, which constructs relative encodings as additional node features to inject the pairwise information into the representation learning process [6, 7, 9, 8]. For example, Souza et al. [8] proposed a relative encoding based on temporal walk counting and theoretically showed that constructing the relative encodings can improve the expressive power of models in distinguishing different links. Despite these advances, existing ways to construct the relative encodings are still far from satisfactory, where computation efficiency is a main concern and temporal information is seldom considered. In this paper, we unify existing relative encodings into a function of temporal walk matrices and explore encoding the pairwise information effectively and efficiently by temporal walk matrix projection.

## 6  Limitation

One limitation of our method is that the matrix construction approach requires manual predefined settings. Different networks may necessitate distinct construction methods, potentially leading to additional human effort in experimenting with various approaches. For instance, the proposed temporal walk matrix that incorporates the time decay effect may not be optimal for networks characterized by long-term dependencies. Developing an adaptive matrix construction technique will be an interesting direction for future research.

## 7  Conclusion

In this paper, we study the problem of pairwise information injection for temporal link prediction. We unify existing construction ways of relative encodings into a unified function, which reveals a connection between the relative encoding and temporal walk matrix. Then we propose a new temporal link prediction model, TPNet, to address the computational inefficiencies and the ignorance of temporal information in previous methods. TPNet introduces a new temporal walk matrix to simultaneously consider the temporal and structural information and a random feature propagation mechanism to maintain the temporal walk matrices efficiently. Theoretically, TPNet preserves the inner product of the maintained temporal walk matrices and empirically outperforms other link-wise methods in both effectiveness and efficiency. An interesting future direction may be designing an adaptive feature propagation mechanism.

## Acknowledgments

This work was supported by the National Natural Science Foundation of China (No. 62272023 and No. 62276015).

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

# A    Analysis of Existing Relative Encodings

## A.1    NAT

In Algorithm 1 of NAT, it maintains a series of hash maps $s_u^{(0)}, s_u^{(1)}, .., s_u^{(k)}$ for each node $u$, which are sets of node ids. Next, we will first prove the statement holds if the size of the hash maps is infinite.

**Statement**. $s^{(i)}$ is a set of nodes that $u$ can approach through a temporal walk whose length is less than $i + 1$.

Initially, the $s_u^{(0)}$ is set as $\{u\}$ and $s_u^{(1)}, .., s_u^{(k)}$ are set as empty set. The statement holds. When a new interaction $(\{u, v\}, t)$ occurs and if the size of the hash maps is infinite, for $1 \le i \le k$, the updating function of hash maps can be represented as

$$\bar{s}_u^{(i)} = s_u^{(i)} \cup s_v^{(i-1)}, \qquad \bar{s}_v^{(i)} = s_v^{(i)} \cup s_u^{(i-1)}, \tag{7}$$

where $\bar{s}_u^{(i)}$ and $s_u^{(i)}$ denote the hash map after and before adding the interaction respectively. Then if the timestamp of $(\{u, v\}, t)$ is larger than that of previous interactions, the newly generated temporal walks beginning from $u$ must first visit $v$ through $(\{u, v\}, t)$ (proved in Section B.1), thus the newly added nodes that $u$ can visit through a temporal walk with length less than $i + 1$ must belong to $s_v^{(i-1)}$. So $\bar{s}_u^{(i)}$ will contain all the nodes that $u$ can approach through a temporal walk with length less than $i + 1$ after adding $(\{u, v\}, t)$. The statement holds.

For a node $w$, its similarity feature $\boldsymbol{r}^{w|u}$ is a $(k + 1)$-dimensional vector, where for $1 \le i \le k + 1$, $r_i^{w|u} = 1$ if $w \in s_u^{(i-1)}$ and $r_i^{w|u} = 0$ otherwise. Considering that the above statement holds, the similarity feature $\boldsymbol{r}^{w|u}$ is equivalent to a $(k + 1)$-dimensional vector, where $r_i^{w|u} = 1$ if the shortest temporal walk from $u$ to $w$ is less than $i$; otherwise, $r_i^{w|u} = 0$.

---

**Algorithm 2:** Temporal Walk Extraction $(\mathcal{G}(t), \alpha, k, u)$

---

1  Initialize $W$ to be $\{(u, t)\}$ ;
2  **for** $i$ *from* $1$ *to* $k$ **do**
3  $\quad$ $(w_{\mathrm{p}}, t_{\mathrm{p}}) \leftarrow$ the last (node, time) pair in $W$;
4  $\quad$ Sample one $(\{w_p, w'\}, t') \in \mathcal{E}_{w_{\mathrm{p}}, t_{\mathrm{p}}}$ with prob. $\propto$
$\quad\quad \exp(-\alpha(t_{\mathrm{p}} - t'))$ ;
5  $\quad$ $W_i \leftarrow W_i \oplus (w', t')$;
6  Return W;

---

## A.2    CAWN

For constructing $\boldsymbol{r}^{w|u}$, CAWN first repeatedly samples $m$ temporal walks of length $k$ begging at $u$ according to the sampling strategy in Algorithm 2. Then $\boldsymbol{r}^{w|u}$ is set to be a $(k + 1)$-dimensional vector, where $r_i^{w|u}$ will be the number of walks whose i-th visited node is $w$ for $1 \le i \le k + 1$, which can be represented as,

$$r_i^{w|u} = \sum_{j=1}^{m} \mathbf{1}_{W_j[i][0]=w}, \tag{8}$$

where $W_j$ is the j-th sampled walks and $\mathbf{1}_{W_j[i][0]=w}$ is 1 if $W_j[i][0] = w$; otherwise, $\mathbf{1}_{W_j[i][0]=w}$ is 0. Due to each temporal walk being sampled independently, $\mathbf{1}_{W_1[i][0]=w}, \cdots, \mathbf{1}_{W_m[i][0]=w}$ are $m$ independent and identically distributed Bernoulli random variables $\mathrm{Ber}(\mu_w^{i-1})$, where $\mu_w^{i-1}$ is the probability of reaching $w$ from $u$ after performing a $(i - 1)$-step temporal walk according to Algorithm 2. And according to the strong law of large numbers (Theorem 1.3.1 of [23]), the mean of these random variables (i.e., $\frac{1}{m} \sum_{j=1}^{m} \mathbf{1}_{W_j[i][0]=w} \iff \frac{1}{m} r_i^{w|u}$) will coverage to the mean as the number of sampled walks $m \to \infty$. The mean of the $\mathrm{Ber}(\mu_w^{i-1})$ is $\mu_w^{i-1}$. Expand all $(i - 1)$-step temporal walks from $u$ to $w$, we have

$$\mu_w^{i-1} = \sum_{W \in \mathcal{M}_{u,w}^{i-1}} f(W), \tag{9}$$

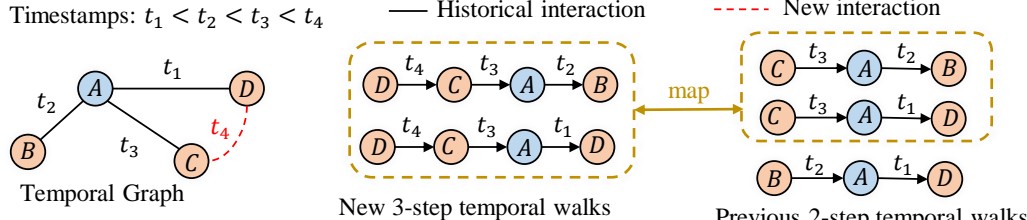

Figure 6: Illustration of the newly generated 3-step temporal walks beginning from $D$ after adding a new interaction $(\{C, D\}, t_4)$ There is a one-to-one map between $\Delta M_{D,*}^3$ and $M_{C,*}^2(t_4)$.

where $f(\cdot)$ is the sampled probability of a walk according to Algorithm 2. For Algorithm 2, if we are current at $(w_i, t_i)$, then the probability of moving to $(w_{i+1}, t_{i+1})$ through an interaction $(\{w_i, w_{i+1}\}, t_{i+1})$ is proportional to $\exp(-\alpha(t_i - t_{i+1}))$, which is $\frac{\exp(-\alpha(t_i - t_{i+1}))}{\sum_{(w', t') \in \mathcal{E}_{w_i, t_i}} \exp(-\alpha(t_i - t'))}$. Thus the probability of a temporal walk $W = [(w_0, t_0), (w_1, t_1), .., (w_k, t_k)]$ is

$$f(W) = \prod_{i=0}^{k-1} \frac{\exp(-\alpha(t_i - t_{i+1}))}{\sum_{(\{w', w\}, t') \in \mathcal{E}_{w_i, t_i}} \exp(-\alpha(t_i - t'))} \tag{10}$$

which is the same as the score function $s'(\cdot)$ of CAWN we shown in Section 2.2.2. Thus, $r_i^{w|u}$ (multiplied with a const $\frac{1}{m}$) is the same as $\sum_{W \in \mathcal{M}_{u,w}^{i-1}} s'(W)$.

## B  Proofs

### B.1  Proof of Theorem 1

The equation in Theorem 1 can be rewritten as $\boldsymbol{H}^{(l)}(t) = e^{\lambda l t} * \boldsymbol{A}^{(l)}(t)\boldsymbol{P}$. We can consider $e^{\lambda l t} * \boldsymbol{A}^{(l)}(t)$ as a new temporal walk matrix whose score function for a temporal walk $[(w_0, t_0), (w_1, t_1), ..., (w_l, t_l)]$ at time $t$ is $e^{\lambda l t} * s(W)$. Expanding it, we have

$$e^{\lambda l t} * s(W) = e^{\lambda l t} * \prod_{j=1}^{l} e^{-\lambda(t - t_j)} = \prod_{j=1}^{l} e^{-\lambda(t - t_j)} * e^{\lambda t} = \prod_{j=1}^{l} e^{\lambda t_j}. \tag{11}$$

We use the notation $\bar{\boldsymbol{A}}^{(l)}(t)$ to denote $e^{\lambda l t} * \boldsymbol{A}^{(l)}(t)$ and $\bar{s}(W)$ to denote $e^{\lambda l t} * s(W)$ for a $l$-step temporal walk in the following proof. The original problem is transformed into proving that $\boldsymbol{H}^{(l)}(t) = \bar{\boldsymbol{A}}^{(l)}(t)\boldsymbol{P}$. Note that for a given temporal walk $[(w_0, t_0), (w_1, t_1), ..., (w_l, t_l)]$, each term of $\bar{s}(W)$ (i.e., $\prod_{j=1}^{l} e^{\lambda t_j}$) will no change as time $t$ goes on. Thus the temporal walk matrices $\bar{\boldsymbol{A}}^{(l)}(t)$ will only change when a new interaction occurs. Next, we inspect how the $\bar{\boldsymbol{A}}^{(l)}(t)$ changes when a new interaction $(u, v, t)$ occurs. For each element $\bar{A}_{i,j}^{(l)}(t)$, its change is caused by the newly generated $l$-step temporal walks from $i$ to $j$ after adding the interaction $(u, v, t)$, which can be written as

$$A_{i,j}^{(l)}(t^+) = A_{i,j}^{(l)}(t) + \sum_{W \in \Delta M_{i,j}^l} \bar{s}(W), \tag{12}$$

where $t^+$ denotes the timestamps right after $t$ and $\Delta M_{i,j}^l$ denote the newly generated $l$-step temporal walks from $i$ to $j$. According to the definition of the temporal walk, the new $l$-step temporal walks must begin from $u$ or $v$. Because if there is a temporal walk that does not begin from $u$ or $v$, it can be represented as $[(w_0, t_0), .., (w_i, t_i), (u, t_{i+1}), (v, t), .., (w_l, t_l)]$ or $[(w_0, t_0), .., (w_i, t_i), (v, t_{i+1}), (u, t), .., (w_l, t_l)]$, which means that there is an interaction $(\{w_i, u\}, t_{i+1})$ or $(\{w_i, v\}, t_{i+1})$ whose timestamp $t_i$ is larger than $t$. (Since the timestamps of a temporal walk are decreasing). This is impossible since $(u, v, t)$ is a newly happened interaction. Thus only the $u$-th row and $v$-th row of the $\bar{\boldsymbol{A}}^{(l)}(t)$ will change. Besides for each newly generated temporal walk from $u$, it must be $[(u, t), (v, t_1), .., (w_l, t_l)]$, where $[(v, t_1), .., (w_l, t_l)]$ corresponds to a $(l-1)$-step temporal walk beginning from $v$. And for any $(l-1)$-step temporal walk beginning

from $v$, we can add a prefix $(u, t)$ to make it become a $l$-step temporal walk beginning from $u$. Thus for any $1 \leq i \leq n$, there is a one-to-one mapping between $\Delta M_{u,i}^l$ and $M_{v,i}^{l-1}(t)$ with $M_{v,i}^{l-1}(t)$ denote the set of $(l-1)$-th temporal walks from $v$ to $i$ before $t$, and we can rewritten Equation (12) as

$$A_{u,i}^{(l)}(t^+) = A_{u,i}^{(l)}(t) + \sum_{W \in M_{v,i}^{l-1}(t)} e^{-\lambda t} * \bar{s}(W) = A_{u,i}^{(l)}(t) + e^{-\lambda t} * A_{v,i}^{(l-1)}(t), \qquad (13)$$

The updating function for $\bar{A}_{v,i}^{(l)}(t)$ is also similar. We give a visual illustration of the new temporal walks in Figure 6 for better understanding. Finally, writing the update formula in vector form, for any $1 \leq l \leq k$ we have

$$\bar{A}_u^{(l)}(t^+) = \bar{A}_u^{(l)}(t) + e^{\lambda t} * \bar{A}_v^{(l-1)}(t), \quad \bar{A}_v^{(l)}(t^+) = \bar{A}_v^{(l)}(t) + e^{\lambda t} * \bar{A}_u^{(l-1)}(t), \qquad (14)$$

which is the same as Equation (14)! Thus if $\boldsymbol{H}^{(l)}(t) = \bar{\boldsymbol{A}}^{(l)}(t)\boldsymbol{P}$ for $0 \leq l \leq k$, after adding a new interaction $(u, v, t)$, for $0 \leq l \leq k$, $\boldsymbol{H}^{(l)}(t^+) = \bar{\boldsymbol{A}}^{(l)}(t^+)\boldsymbol{P}$ still holds. Because we have

$$\boldsymbol{H}_u^{(l)}(t^+) = \boldsymbol{H}_u^{(l)}(t) + e^{\lambda t} * \boldsymbol{H}_v^{(l-1)}(t) = (\bar{\boldsymbol{A}}_u^{(l)}(t) + e^{\lambda t} * \bar{\boldsymbol{A}}_v^{(l-1)}(t))\boldsymbol{P} = \bar{\boldsymbol{A}}_u^{(l)}(t^+)\boldsymbol{P} \qquad (15)$$

Note that the equation in Equation (1) holds at initialization, thus the equation always holds and the theorem is proved.

## B.2 Proof of Theorem 2

The Theorem 2 can be considered as a special case of the Johnson-Lindenstrauss Lemma [11], where the random projection [12, 24, 25] can preserve the norm and inner product. For the convenience of readers lacking relevant background, we follow [26] to provide the proof under the given conditions of this paper. We begin the proof by the following lemma.

**Lemma 1** ((Lemma 3.18 of [26])). *Let $\boldsymbol{x} \in \mathbb{R}^d$ be a d-dimensional random vector whose entries are independent $\mathcal{N}(0, \frac{1}{d})$. Then for any $\epsilon \in [0, 1]$,*

$$\mathbb{P}\left(\left|\|\boldsymbol{x}\|_2^2 - 1\right| > \epsilon\right) \leq 2\exp(\frac{-\epsilon^2 d}{8}) \qquad (16)$$

The following part can be divided into 1) We first give two corollaries and their proofs. 2) Then we give the proof of Theorem 2 based on the corollaries.

### B.2.1 Two Corolarries

Based on the above lemma, we can get the following corollaries.

**Corollary 1.** *Given any $\epsilon \in (0, 1), \boldsymbol{x} \in \mathbb{R}^m$. Let $\boldsymbol{P} \in \mathbb{R}^{d \times m}$ be a random matrix whose entries are independent $\mathcal{N}(0, \frac{1}{d})$, if $d \geq \frac{8}{\epsilon^2}\log(\frac{1}{\delta})$, we have*

$$\mathbb{P}\left((1-\epsilon)\|\boldsymbol{x}\|_2^2 \leq \|\boldsymbol{P}\boldsymbol{x}\|_2^2 \leq (1+\epsilon)\|\boldsymbol{x}\|_2^2\right) \geq 1 - 2\delta \qquad (17)$$

**Proof**. For the above corollary, we have

$$\mathbb{P}\left((1-\epsilon)\|\boldsymbol{x}\|_2^2 \leq \|\boldsymbol{P}\boldsymbol{x}\|_2^2 \leq (1+\epsilon)\|\boldsymbol{x}\|_2^2\right) \geq 1 - 2\delta$$
$$\iff \mathbb{P}\left(\left|\frac{\|\boldsymbol{P}\boldsymbol{x}\|_2^2}{\|\boldsymbol{x}\|_2^2} - 1\right| \leq \epsilon\right) \geq 1 - 2\delta \qquad (18)$$
$$\iff \mathbb{P}\left(\left|\frac{\|\boldsymbol{P}\boldsymbol{x}\|_2^2}{\|\boldsymbol{x}\|_2^2} - 1\right| > \epsilon\right) \leq 2\delta$$

Note that each entry of $\boldsymbol{P}$ is an independent $\mathcal{N}(0, \frac{1}{d})$, thus $\boldsymbol{P}\boldsymbol{x}$ is a random vector whose each entry $(\boldsymbol{P}\boldsymbol{x})_k = \sum_{i=1}^m P_{k,i} * x_i$ is an independent $\mathcal{N}(0, \frac{\|\boldsymbol{x}\|_2^2}{d})$ and each entry of $\frac{\boldsymbol{P}\boldsymbol{x}}{\|\boldsymbol{x}\|_2}$ is an independent $\mathcal{N}(0, \frac{1}{d})$. Substituting it to Lemma 1 and taking $d \geq \frac{8}{\epsilon^2}\log(\frac{1}{\delta})$, we have

$$\mathbb{P}\left(\left|\frac{\|\boldsymbol{P}\boldsymbol{x}\|_2^2}{\|\boldsymbol{x}\|_2^2} - 1\right| > \epsilon\right) \leq 2\exp(\frac{-\epsilon^2 d}{8})$$
$$\leq 2\exp(\frac{-\epsilon^2}{8} * \frac{8}{\epsilon^2} * \log(\frac{1}{\delta})) \leq 2\delta \qquad (19)$$

Based on Corollary 1, we can get the following corollary.

**Corollary 2.** *Given any $n$ $m$-dimensional vectors $\boldsymbol{x}_1, ..., \boldsymbol{x}_n$, $\epsilon \in (0,1)$, let $\boldsymbol{P} \in \mathbb{R}^{d \times m}$ be a random matrix whose entries are independent $\mathcal{N}(0, \frac{1}{d})$, if $d \geq \frac{24}{\epsilon^2} \log(4^{1/3}n)$, for any $1 \leq i, j \leq n$, we have*

$$\mathbb{P}\left(|\langle \boldsymbol{P}\boldsymbol{x_i}, \boldsymbol{P}\boldsymbol{x_j}\rangle - \langle \boldsymbol{x_i}, \boldsymbol{x_j}\rangle| \leq \frac{\epsilon}{2}(\|\boldsymbol{x}_i\|_2^2 + \|\boldsymbol{x}_j\|_2^2)\right) \geq 1 - \frac{1}{n} \tag{20}$$

let $E_{i,j}^+$ and $E_{i,j}^-$ denote the event of $\{|\|\boldsymbol{P}(\boldsymbol{x}_i + \boldsymbol{x}_j)\|_2^2 - \|\boldsymbol{x}_i + \boldsymbol{x}_j\|_2^2| \leq \epsilon\|\boldsymbol{x}_i + \boldsymbol{x}_j\|_2^2\}$ and $\{|\|\boldsymbol{P}(\boldsymbol{x_i} - \boldsymbol{x_j})\|_2^2 - \|\boldsymbol{x}_i - \boldsymbol{x}_j\|_2^2| \leq \epsilon\|\boldsymbol{x}_i - \boldsymbol{x}_j\|_2^2\}$ respectively. Since $\boldsymbol{x}_i + \boldsymbol{x}_j$ and $\boldsymbol{x}_i - \boldsymbol{x}_j$ can also be consider two $m$-dimensional vectors. Thus take it and $d \geq \frac{24}{\epsilon^2} \log(4^{1/3}n)$ into Corollary 1, we have $\mathbb{P}(E_{i,j}^+) \geq 1 - \frac{1}{2n^3}$ and $\mathbb{P}(E_{i,j}^-) \geq 1 - \frac{1}{2n^3}$. Then let $C_{i,j} = E_{i,j}^+ \cap E_{i,j}^-$ denote that $E_{i,j}^+$ and $E_{i,j}^-$ hold simultaneously, according to the union bound, we have

$$\mathbb{P}(C_{i,j}) = 1 - \mathbb{P}(\overline{E_{i,j}^+} \cup \overline{E_{i,j}^-}) \geq 1 - (\mathbb{P}(\overline{E_{i,j}^+}) + \mathbb{P}(\overline{E_{i,j}^-})) \geq 1 - \frac{1}{n^3}, \tag{21}$$

where $\overline{E_{i,j}^+}$ denote that $E_{i,j}^+$ does not hold and $\overline{E_{i,j}^+} \cup \overline{E_{i,j}^-}$ denotes that $\overline{E_{i,j}^+}$ or $\overline{E_{i,j}^-}$ holds. According to the union bound, we further have

$$\mathbb{P}(\cap_{i,j} C_{i,j}) = 1 - \mathbb{P}(\cup_{i,j}\overline{C_{i,j}}) \geq 1 - \sum_{i,j} \mathbb{P}(\overline{C_{i,j}}) \geq 1 - n^2 * \frac{1}{n^3} \geq 1 - \frac{1}{n}, \tag{22}$$

where $\cap_{i,j}C_{i,j}$ denote that $C_{i,j}$ holds for any $i, j$ and $\cup_{i,j}\overline{C_{i,j}}$ denote that there exist $i, j$ that $\overline{C_{i,j}}$ holds. If $C_{i,j}$ holds, we have

$$(1 - \epsilon)\|\boldsymbol{x_i} + \boldsymbol{x_j}\|_2^2 \leq \|\boldsymbol{P}(\boldsymbol{x_i} + \boldsymbol{x_j})\|_2^2 \leq (1 + \epsilon)\|\boldsymbol{x_i} + \boldsymbol{x_j}\|_2^2 \tag{23}$$

$$(1 - \epsilon)\|\boldsymbol{x_i} - \boldsymbol{x_j}\|_2^2 \leq \|\boldsymbol{P}(\boldsymbol{x_i} - \boldsymbol{x_j})\|_2^2 \leq (1 + \epsilon)\|\boldsymbol{x_i} - \boldsymbol{x_j}\|_2^2 \tag{24}$$

Multiplying Equation (24) with $-1$ and adding it to (23), we have

$$|\langle \boldsymbol{P}\boldsymbol{x_i}, \boldsymbol{P}\boldsymbol{x_j}\rangle - \langle \boldsymbol{x_i}, \boldsymbol{x_j}\rangle| \leq \frac{\epsilon}{2}(\|\boldsymbol{x_i}\|_2^2 + \|\boldsymbol{x_j}\|_2^2) \tag{25}$$

Thus we have

$$\mathbb{P}(\cap_{i,j}C_{i,j}) \geq 1 - \frac{1}{n} \implies \mathbb{P}(|\langle \boldsymbol{P}\boldsymbol{x_i}, \boldsymbol{P}\boldsymbol{x_j}\rangle - \langle \boldsymbol{x_i}, \boldsymbol{x_j}\rangle| \leq \frac{\epsilon}{2}(\|\boldsymbol{x_i}\|_2^2 + \|\boldsymbol{x_j}\|_2^2)) \geq 1 - \frac{1}{n} \tag{26}$$

holds for any $1 \leq i, j \leq n$.

### B.2.2 Proof based on Corollaries

The matrix of $\boldsymbol{A}^{(l)}(t)$ in Theorem 2 can be considered as $n$ vectors with $n$ dimensions, where each row of $\boldsymbol{A}^{(l)}(t)$ is a $n$-dimensional vector. Similarly, we can consider $\boldsymbol{A}^{(0)}(t), \cdots, \boldsymbol{A}^{(k)}(t)$ together as $n(k+1)$ vectors with $n$ dimensions. Considering that $\boldsymbol{P} \in \mathbb{R}^{n \times d_R}$ is a random matrix where each entry is an independent $\mathcal{N}(0, \frac{1}{d_R})$ and $e^{-\lambda l t} * \boldsymbol{H}^{(l)}(t)$ is the projection of $\boldsymbol{A}^{(l)}(t)$, substitute it into Corollary 2 and taking the number of vectors as $(k+1)n$, we can get Theorem 2.

## C  Batch Updating Mechanism

For the situation where multiple interactions happen simultaneously, we can first compute each interaction's contribution independently and sum them together to update the node representations. The maintaining mechanism is shown in Algorithm 3, where we packed the interactions that happen at the same time into a set $\mathcal{B}$ and sum the independent contribution of each interaction in $\mathcal{B}$ into $\Delta\boldsymbol{H}^{(1)}, \cdots, \Delta\boldsymbol{H}^{(k)}$. For simplicity, we initialize $\Delta\boldsymbol{H}^{(1)}, \cdots, \Delta\boldsymbol{H}^{(k)}$ each time and it can be replaced by some efficient implementation such as scatter_add operation in pytorch.

**Algorithm 3:** Batch Node Representation Maintaining $(\mathcal{G}, \lambda, k, n, d_R)$

---

1   Initialize $\boldsymbol{H}^{(0)}, \boldsymbol{H}^{(1)}, ..., \boldsymbol{H}^{(k)} \in \mathbb{R}^{n \times d_R}$ as zero matrix ;

2   Fill $\boldsymbol{H}^{(0)}$ with entries independently drawn from $\mathcal{N}(0, \frac{1}{d_R})$ ;

3   **for** $\mathcal{B} \in \mathcal{G}$ **do**

4      Initialize $\Delta\boldsymbol{H}^{(1)}, ..., \Delta\boldsymbol{H}^{(k)} \in \mathbb{R}^{n \times d_R}$ as zero matrix ;

5      **for** $(u, v, t) \in \mathcal{B}$ **do**

6         **for** $l = k$ **to** 1 **do**

7            $\Delta\boldsymbol{H}_u^{(l)} = \Delta\boldsymbol{H}_u^{(l)} + e^{\lambda t} * \boldsymbol{H}_v^{(l-1)}$ ;

8            $\Delta\boldsymbol{H}_v^{(l)} = \Delta\boldsymbol{H}_v^{(l)} + e^{\lambda t} * \boldsymbol{H}_u^{(l-1)}$ ;

9      **for** $l = k$ **to** 1 **do**

10         $\boldsymbol{H}^{(l)} = \boldsymbol{H}^{(l)} + \Delta\boldsymbol{H}^{(l)}$ ;

11 Return $\boldsymbol{H}^{(0)}, \boldsymbol{H}^{(1)}, .., \boldsymbol{H}^{(k)}$ ;

---

## D   Propagation Mechanism for Other Matrices

For simplicity, here we only consider the situation where the timestamp of each interaction is different and we can adopt a similar batch updating mechanism in Section C to handle the situation where multiple interactions happen simultaneously. For the updating of other types of temporal walk matrices, let $\boldsymbol{A}(t) = [\boldsymbol{A}^{(0)}(t), \cdots, \boldsymbol{A}^{(k)}(t)] \in \mathbb{R}^{n(k+1) \times d}$ denotes the concatenation of the temporal walk matrices. If the following two situations can be satisfied simultaneously, we can apply the random feature propagation mechanism to implicitly maintain the temporal walk matrices.

**Condition 1**. After adding a new interaction $(u, v, t)$, the change of each row can be written as the linear combination of other rows, which is for each $1 \le i \le n(k+1)$, there exists $k_1, ..., k_m$ and $l_1, ..., l_m$ satisfying.

$$\boldsymbol{A}_i(t^+) = k_1 \boldsymbol{A}_{l_1}(t) + \cdots + k_m \boldsymbol{A}_{l_m}(t), \tag{27}$$

where $t^+$ is the time right after $t$.

**Condition 2**. After time moving $\Delta t$ without adding new interactions, the change of each row can be written as the linear combination of other rows, which is for each $1 \le i \le n(k+1)$, there exists $k_1, ..., k_m$ and $l_1, ..., l_m$ satisfying.

$$\boldsymbol{A}_i(t + \Delta t) = k_1 \boldsymbol{A}_{l_1}(t) + \cdots + k_m \boldsymbol{A}_{l_m}(t) \tag{28}$$

The motivation behind the conditions is that the projection is a linear operation. If the node representations are the projection of the temporal walk matrix at $t$ (i.e., $\boldsymbol{H}(t) = \boldsymbol{A}(t)\boldsymbol{P}$) and the updating function of the temporal walk matrix is the linear combination of other rows, then we can apply the same updating function on $\boldsymbol{H}(t)$, which will make $\boldsymbol{H}(t^+)$ still be the projection of the temporal walk matrix. For example, applying (27) to $\boldsymbol{H}(t)$ will get

$$\begin{aligned} \boldsymbol{H}_i(t^+) &= k_1 \boldsymbol{H}_{l_1}(t) + \cdots + k_m \boldsymbol{H}_{l_m}(t) \\ &= (k_1 \boldsymbol{A}_{l_1}(t) + \cdots + k_m \boldsymbol{A}_{l_m}(t))\boldsymbol{P} = \boldsymbol{A}_i(t^+)\boldsymbol{P} \end{aligned} \tag{29}$$

Then we can ensure that $\boldsymbol{H}(t)$ is always the random projection of $\boldsymbol{A}(t)$ and thus preserve the inner product of the $\boldsymbol{A}(t)$.

### D.1   Detailed Updating Mechanism

Based on the above analysis, we give the detailed propagation mechanism of methods in Section 2. For NAT, PINT and DyGFormer, their temporal matrix element $A_{u,v}^{(l)}(t)$ is the number of the $l$-step temporal walks from $u$ to $v$ and their feature propagation mechanism is shown in Algorithm 4, where the obtained node representations are the projection of the corresponding temporal walk matrix.

For CAWN, its score function for a temporal walk $W = [(w_0, t_0), (w_1, t_1), ..., (w_l, t_l)]$ is defined as $s(W) = \prod_{i=0}^{l-1} \frac{\exp(-\alpha(t_i - t_{i+1}))}{\sum_{(\{w', w\}, t') \in \mathcal{E}_{w_i, t_i}} \exp(-\alpha(t_i - t'))}$. As time goes on, its element of temporal walk

---
**Algorithm 4:** Propagation Mechanism for DyGFormer ($\mathcal{G}$,$k$,$n$,$d_R$)
---
1   Initialize $\boldsymbol{H}^{(0)}, \boldsymbol{H}^{(1)}, ..., \boldsymbol{H}^{(k)} \in \mathbb{R}^{n \times d_R}$ as zero matrix ;
2   Fill $\boldsymbol{H}^{(0)}$ with entries independently drawn from $\mathcal{N}(0, \frac{1}{d_R})$ ;
3   **for** $(u, v, t) \in \mathcal{G}$ **do**
4      **for** $l = k$ **to** 1 **do**
5         $\boldsymbol{H}_u^{(l)} = \boldsymbol{H}_u^{(l)} + \boldsymbol{H}_v^{(l-1)}$ ;
6         $\boldsymbol{H}_v^{(l)} = \boldsymbol{H}_v^{(l)} + \boldsymbol{H}_u^{(l-1)}$ ;
7   Return $\boldsymbol{H}^{(0)}, \boldsymbol{H}^{(1)}, .., \boldsymbol{H}^{(k)}$;
---

---
**Algorithm 5:** Propagation Mechanism for CAWN ($\mathcal{G}$,$\alpha$,$k$,$n$,$d_R$)
---
1   Initialize $\boldsymbol{H}^{(0)}, \boldsymbol{H}^{(1)}, ..., \boldsymbol{H}^{(k)} \in \mathbb{R}^{n \times d_R}$ as zero matrix ;
2   Fill $\boldsymbol{H}^{(0)}$ with entries independently drawn from $\mathcal{N}(0, \frac{1}{d_R})$ ;
3   Initialize $\boldsymbol{d} \in \mathbb{R}^n$ as zero vector ;
4   Set $t_{\text{prev}} \in \mathbb{R}$ to be zero ;
5   **for** $(u, v, t) \in \mathcal{G}$ **do**
6      $\boldsymbol{d} = \boldsymbol{d} * \exp(-\alpha(t - t_{\text{prev}}))$ ;
7      **for** $l = k$ **to** 1 **do**
8         $\boldsymbol{H}_u^{(l)} = \frac{d_u}{d_u+1} * \boldsymbol{H}_u^{(l)} + \frac{1}{d_u+1} * \boldsymbol{H}_v^{(l-1)}$ ;
9         $\boldsymbol{H}_v^{(l)} = \frac{d_v}{d_v+1} * \boldsymbol{H}_v^{(l)} + \frac{1}{d_v+1} * \boldsymbol{H}_u^{(l-1)}$ ;
10     $d_u = d_u + 1$ ;
11     $d_v = d_v + 1$ ;
12     $t_{\text{prev}} = t$ ;
13   Return $\boldsymbol{H}^{(0)}, \boldsymbol{H}^{(1)}, .., \boldsymbol{H}^{(k)}$;
---

matrices will not change. When a new interaction $(u, v, t)$ happens, for a temporal walk matrix $\boldsymbol{A}^{(l)}(t)$, its $u$-th row and $v$-th row will change. Formally, let $\boldsymbol{d}(t) \in \mathbb{R}^n$ denotes a time decay degree vector, where for each node $u$, $d_u(t)$ is defined as $d_u(t) = \sum_{(\{u,v'\},t') \in \mathcal{E}_{u,t}} \exp(-\alpha(t' - t))$ with $\mathcal{E}_{u,t}$ denoting the set of interactions attached to $u$ before $t$, then the updating function of the temporal walk matrix $\boldsymbol{A}^{(k)}(t)$ can be represented as

$$
\begin{aligned}
\boldsymbol{A}_u^{(l)}(t^+) &= \frac{d_u(t)}{d_u(t) + 1} * \boldsymbol{A}_u^{(l)}(t) + \frac{1}{d_u(t) + 1} * \boldsymbol{A}_v^{(l-1)}(t), \\
\boldsymbol{A}_v^{(l)}(t^+) &= \frac{d_v(t)}{d_v(t) + 1} * \boldsymbol{A}_v^{(l)}(t) + \frac{1}{d_v(t) + 1} * \boldsymbol{A}_u^{(l-1)}(t),
\end{aligned}
\tag{30}
$$

where $\frac{d_u(t)}{d_u(t)+1} * \boldsymbol{A}_u^{(l)}(t)$ corresponds to the change in score of the old $l$-step temporal walks begging from $u$ and $\frac{1}{d_u(t)+1} * \boldsymbol{A}_v^{(l-1)}(t)$ correspond to change caused by the newly generated $l$-step temporal walk from $u$ (the same for $v$ and similar analysis about the updating of temporal walk matrix can be found in Appendix B.1). Then we can give the propagation mechanism for CAWN in Algorithm 5 based on the above analysis, where $\boldsymbol{d}$ corresponds to the time decay degree vector.

## E   Broader Impact

We proposed an effective and efficient temporal link prediction method, which may advance real-world scenarios that rely on link prediction as a cornerstone, such as recommendation systems. For potential negative impacts, overly accurate link prediction in some contexts may lead to imbalanced outcomes such as reduced diversity in recommendation systems.

# F  Experimental Settings

## F.1  Datasets

Experiments are conducted on the following 13 benchmark datasets [3] collected by [16].

- **Wikipedia** is a bipartite interaction graph that records the edits on Wikipedia pages over a month. Nodes represent users and pages, and links denote the editing behaviors with timestamps. Each link is associated with a 172-dimensional Linguistic Inquiry and Word Count (LIWC) feature [27].
- **Reddit** is a bipartite graph capturing user posts under subreddits over a month. Nodes represent users and subreddits, while links denote timestamped posts. Each link carries a 172-dimensional LIWC feature.
- **MOOC** is a bipartite interaction network of online courses, where nodes represent students and course content units (e.g., videos, problem sets). Links indicate students' access to specific content units and have a 4-dimensional feature.
- **LastFM** is a bipartite network detailing song-listening behaviors of users over one month. Nodes are users and songs, and links represent listening activities.
- **Enron** records email communications between employees of the Enron Energy Corporation over three years.
- **Social Evo.** is a mobile phone proximity network tracking daily activities within an undergraduate dormitory for eight months, with each link having a 2-dimensional feature.
- **UCI** is an online communication network with nodes representing university students and links representing posted messages.
- **Flights** is a dynamic flight network showing air traffic development during the COVID-19 pandemic. Nodes represent airports, and links denote tracked flights. Each link has a weight indicating the number of flights between two airports per day.
- **Can. Parl.** is a dynamic political network recording interactions between Canadian Members of Parliament (MPs) from 2006 to 2019. Nodes represent MPs from electoral districts, and links are formed when two MPs vote "yes" on a bill. The weight of each link represents the number of times one MP voted "yes" in support of another MP in a year.
- **US Legis.** is a senate co-sponsorship network tracking social interactions between US Senators. The weight of each link indicates the number of times two senators co-sponsored a bill in a given congress.
- **UN Trade** captures food and agriculture trade between 181 nations over more than 30 years. The weight of each link represents the total normalized agriculture import or export values between two countries.
- **UN Vote** records roll-call votes in the United Nations General Assembly. A link between two nations increases in weight each time both vote "yes" on an item.
- **Contact** tracks the evolution of physical proximity among approximately 700 university students over a month. Each student has a unique identifier, and links indicate close proximity, with weights revealing the extent of physical closeness between students.

The statistics of the datasets are shown in Table 3, where #N&L Feat stands for the dimensions of node and link features.

## F.2  Baselines

We select the following eleven popular baselines:

- **JODIE** [3] designs a recurrent architecture to maintain a memory vector for each node and a projection layer to map the node memories into future representation trajectories.
- **DyRep** [28] considers each link as a temporal point process [29] and designs a deep temporal point process model to capture the dynamics of the observed process.

---

[3] https://zenodo.org/record/7213796#.Y1cO6y8r30o

Table 3: Statistics of the datasets.

| Datasets | Domains | #Nodes | #Links | #N&L Feat | Bipartite | Duration | Unique Steps | Time Granularity |
|---|---|---|---|---|---|---|---|---|
| Wikipedia | Social | 9,227 | 157,474 | – & 172 | True | 1 month | 152,757 | Unix timestamps |
| Reddit | Social | 10,984 | 672,447 | – & 172 | True | 1 month | 669,065 | Unix timestamps |
| MOOC | Interaction | 7,144 | 411,749 | – & 4 | True | 17 months | 345,600 | Unix timestamps |
| LastFM | Interaction | 1,980 | 1,293,103 | – & – | True | 1 month | 1,283,614 | Unix timestamps |
| Enron | Social | 184 | 125,235 | – & – | False | 3 years | 22,632 | Unix timestamps |
| Social Evo. | Proximity | 74 | 2,099,519 | – & 2 | False | 8 months | 565,932 | Unix timestamps |
| UCI | Social | 1,899 | 59,835 | – & – | False | 196 days | 58,911 | Unix timestamps |
| Flights | Transport | 13,169 | 1,927,145 | – & 1 | False | 4 months | 122 | days |
| Can. Parl. | Politics | 734 | 74,478 | – & 1 | False | 14 years | 14 | years |
| US Legis. | Politics | 225 | 60,396 | – & 1 | False | 12 congresses | 12 | congresses |
| UN Trade | Economics | 255 | 507,497 | – & 1 | False | 32 years | 32 | years |
| UN Vote | Politics | 201 | 1,035,742 | – & 1 | False | 72 years | 72 | years |
| Contact | Proximity | 692 | 2,426,279 | – & 1 | False | 1 month | 8,064 | 5 minutes |

- **TGAT** [14] proposes a time encoding function based on Bochner's Theorem [30] and combines it with the graph attention mechanism [31] to learn dynamic node representations.

- **TGN** [21] proposes a general framework for temporal graph learning, which includes a memory module to maintain node memories and an embedding module to aggregate node memories.

- **CAWN** [6] captures the evolution pattern of the temporal graph by causal anonymous walks. For a given link, CAWN first sampled a set of temporal walks beginning from the two end nodes respectively and constructs relative encodings. The sampled walks together with relative encodings are then mapped into node representations via a sequential model.

- **EdgeBank** [16] is a statistical method, that gives the likelihood of a link based on historical interactions between the two end nodes.

- **TCL** [32] propose a graph transformer architecture [33] to learning node representations, which samples neighbor nodes based on BFS and maps them into the node representation.

- **NAT** [7] propose a dictionary-type neighborhood representation to efficiently capture the correlation information between nodes, which maintains a series of N-caches to store the neighborhood information and use them to decode the pairwise information between nodes.

- **PINT** [8] proposes an injective temporal message passing mechanism to learn node representations and relative positional features constructed based on temporal walk counting to inject pairwise information between nodes.

- **GraphMixer** [13] proposes a simplified temporal graph learning architecture, which employs MLP-Mixer to learn the representation of a node from its historical interaction sequences.

- **DyGFormer** [9] proposes a transformer architecture [34] to learning node representations, which include a patching technique to capture the long-term histories of a node to and a neighbor co-occurrence encoding scheme to capture the correlation between nodes.

## G   Additional Experimental Results

### G.1   Performance Comparison

The results under other settings are shown in Table 6, Table 7, Table 8, Table 9 and Table 10.

### G.2   Efficiency Analysis

Efficiency analysis results on Reddit, UCI, and Wikipedia are shown in Figure 7.

### G.3   Scalability Analysis

The scalability analysis result of PINT is shown in Table 4, where OOM indicates the out-of-memory error.

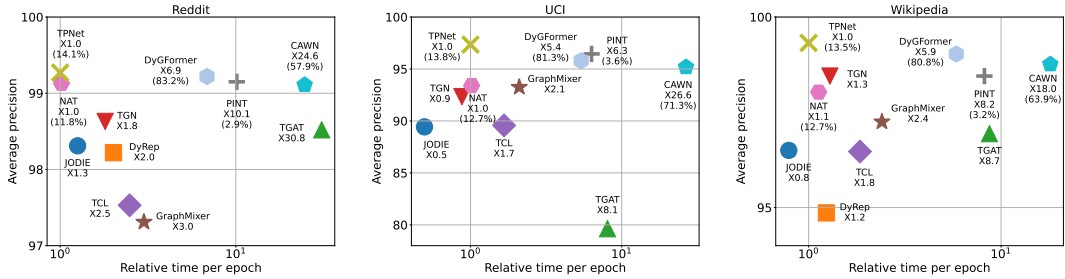

Figure 7: Efficiency analysis on more datasets.

Table 4: Scalability analysis of PINT.

| Edge Number | PINT | |
| --- | --- | --- |
| | Time | Memory |
| 100000 | 51.18 | 92.32 |
| 1000000 | 594.77 | 2175.80 |
| 10000000 | N/A | OOM |
| 100000000 | N/A | OOM |

Table 5: Average norm of node representations from different layers.

| Dataset | Layer1 | Layer2 | Layer3 |
| --- | --- | --- | --- |
| Wikipedia | 1.16E+01 | 1.23E+03 | 2.06E+05 |
| Reddit | 4.28E+01 | 2.52E+05 | 2.54E+09 |
| MOOC | 5.70E+00 | 2.50E+03 | 1.52E+06 |
| LastFM | 9.64E+01 | 1.81E+04 | 4.78E+06 |
| Enron | 2.73E-01 | 1.50E+00 | 4.60E-02 |
| Social Evo. | 2.81E+03 | 7.30E+06 | 1.56E+10 |
| UCI | 1.88E+01 | 8.67E+02 | 6.27E+04 |
| Flights | 3.38E+01 | 8.12E+03 | 3.29E+06 |
| Can. Parl. | 8.36E+00 | 7.49E+02 | 1.40E+04 |
| US Legis. | 4.39E+01 | 2.75E+03 | 1.22E+05 |
| UN Trade | 1.67E+02 | 1.58E+04 | 1.04E+06 |
| UN Vote | 2.41E+02 | 3.36E+04 | 3.22E+06 |
| Contact | 9.88E+00 | 8.83E+02 | 9.36E+04 |

### G.4 Influence of Node Representation Dimension

Results on Wikipedia, Enron, and UCI are shown in Figure 8.

### G.5 Statistic Analysis of Node Representations

Table 5 shows the mean of node representation norm at different layers, where aEb indicates $a \times 10^b$.

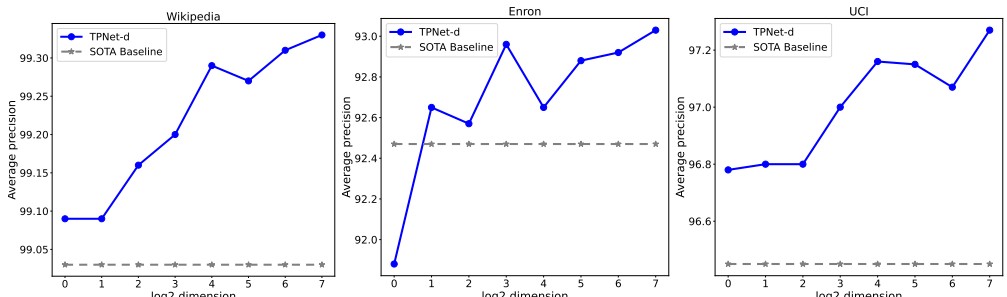

Figure 8: Influence of Node Representation Dimension

Table 6: Inductive results for random negative sampling.

| Metrics | Datasets | JODIE | DyRep | TGAT | TGN | CAWN | TCL | GraphMixer | NAT | PINT | DyGFormer | RPNet |
|---------|----------|-------|-------|------|-----|------|-----|------------|-----|------|-----------|-------|
| AP | Wikipedia | $94.82_{\pm0.20}$ | $92.43_{\pm0.37}$ | $96.22_{\pm0.07}$ | $97.83_{\pm0.04}$ | $98.24_{\pm0.03}$ | $96.22_{\pm0.17}$ | $96.65_{\pm0.02}$ | $96.30_{\pm0.08}$ | $98.03_{\pm0.04}$ | $\underline{98.59_{\pm0.03}}$ | $\mathbf{98.91_{\pm0.01}}$ |
| | Reddit | $96.50_{\pm0.13}$ | $96.09_{\pm0.11}$ | $97.09_{\pm0.04}$ | $97.50_{\pm0.07}$ | $98.62_{\pm0.01}$ | $94.09_{\pm0.07}$ | $95.26_{\pm0.02}$ | $98.24_{\pm0.04}$ | $98.56_{\pm0.05}$ | $\underline{98.84_{\pm0.02}}$ | $\mathbf{98.86_{\pm0.01}}$ |
| | MOOC | $79.63_{\pm1.92}$ | $81.07_{\pm0.44}$ | $85.50_{\pm0.19}$ | $\underline{89.04_{\pm1.17}}$ | $81.42_{\pm0.24}$ | $80.60_{\pm0.22}$ | $81.41_{\pm0.21}$ | $83.62_{\pm1.19}$ | $87.90_{\pm0.98}$ | $86.96_{\pm0.43}$ | $\mathbf{95.07_{\pm0.26}}$ |
| | LastFM | $81.61_{\pm3.82}$ | $83.02_{\pm1.48}$ | $78.63_{\pm0.31}$ | $81.45_{\pm4.29}$ | $89.42_{\pm0.07}$ | $73.53_{\pm1.66}$ | $82.11_{\pm0.42}$ | $92.24_{\pm0.93}$ | $92.42_{\pm0.64}$ | $\underline{94.23_{\pm0.09}}$ | $\mathbf{95.36_{\pm0.11}}$ |
| | Enron | $80.72_{\pm1.39}$ | $74.55_{\pm3.95}$ | $67.05_{\pm1.51}$ | $77.94_{\pm1.02}$ | $86.35_{\pm0.51}$ | $76.14_{\pm0.79}$ | $75.88_{\pm0.48}$ | $87.18_{\pm1.24}$ | $88.12_{\pm0.30}$ | $\underline{89.76_{\pm0.34}}$ | $\mathbf{90.34_{\pm0.28}}$ |
| | Social Evo. | $91.96_{\pm0.48}$ | $90.04_{\pm0.47}$ | $91.41_{\pm0.16}$ | $90.77_{\pm0.86}$ | $79.94_{\pm0.18}$ | $91.55_{\pm0.09}$ | $91.86_{\pm0.06}$ | $87.44_{\pm5.48}$ | $92.40_{\pm0.60}$ | $\underline{93.14_{\pm0.04}}$ | $\mathbf{93.24_{\pm0.07}}$ |
| | UCI | $79.86_{\pm1.48}$ | $57.48_{\pm1.87}$ | $79.54_{\pm0.48}$ | $88.12_{\pm2.05}$ | $92.73_{\pm0.06}$ | $87.36_{\pm2.03}$ | $91.19_{\pm0.42}$ | $87.31_{\pm0.28}$ | $\underline{94.72_{\pm0.15}}$ | $94.54_{\pm0.12}$ | $\mathbf{95.74_{\pm0.05}}$ |
| | Flights | $94.74_{\pm0.37}$ | $92.88_{\pm0.73}$ | $88.73_{\pm0.33}$ | $95.03_{\pm0.60}$ | $97.06_{\pm0.02}$ | $83.41_{\pm0.07}$ | $83.03_{\pm0.05}$ | $96.74_{\pm0.22}$ | $97.54_{\pm0.06}$ | $\underline{97.79_{\pm0.02}}$ | $\mathbf{97.97_{\pm0.04}}$ |
| | Can. Parl. | $53.92_{\pm0.94}$ | $54.02_{\pm0.76}$ | $55.18_{\pm0.79}$ | $54.10_{\pm0.93}$ | $55.80_{\pm0.69}$ | $54.30_{\pm0.66}$ | $55.91_{\pm0.82}$ | $61.90_{\pm2.52}$ | $50.32_{\pm0.86}$ | $\mathbf{87.74_{\pm0.71}}$ | $\underline{68.09_{\pm1.55}}$ |
| | US Legis. | $54.93_{\pm2.29}$ | $57.28_{\pm0.71}$ | $51.00_{\pm3.11}$ | $58.63_{\pm0.37}$ | $53.17_{\pm1.20}$ | $52.59_{\pm0.97}$ | $50.71_{\pm0.76}$ | $\underline{60.41_{\pm0.74}}$ | $59.71_{\pm1.36}$ | $54.28_{\pm2.87}$ | $\mathbf{61.71_{\pm0.84}}$ |
| | UN Trade | $59.65_{\pm0.77}$ | $57.02_{\pm0.69}$ | $61.03_{\pm0.18}$ | $58.31_{\pm3.15}$ | $65.24_{\pm0.21}$ | $62.21_{\pm0.12}$ | $62.17_{\pm0.31}$ | $\underline{69.57_{\pm1.45}}$ | $60.37_{\pm0.78}$ | $64.55_{\pm0.62}$ | $\mathbf{86.53_{\pm0.29}}$ |
| | UN Vote | $56.64_{\pm0.96}$ | $54.62_{\pm2.22}$ | $52.24_{\pm1.46}$ | $\underline{58.85_{\pm2.51}}$ | $49.94_{\pm0.45}$ | $51.60_{\pm0.97}$ | $50.68_{\pm0.44}$ | $\mathbf{66.60_{\pm0.98}}$ | $57.43_{\pm1.24}$ | $55.93_{\pm0.39}$ | $58.00_{\pm3.21}$ |
| | Contact | $94.34_{\pm1.45}$ | $92.18_{\pm0.41}$ | $95.87_{\pm0.11}$ | $93.82_{\pm0.99}$ | $89.55_{\pm0.30}$ | $91.11_{\pm0.12}$ | $90.59_{\pm0.05}$ | $96.12_{\pm0.08}$ | $97.41_{\pm0.14}$ | $\underline{98.03_{\pm0.02}}$ | $\mathbf{98.39_{\pm0.02}}$ |
| | Avg. Rank | 7.62 | 8.69 | 7.92 | 6.15 | 6.15 | 8.46 | 7.85 | 4.69 | 4.15 | $\underline{3.00}$ | **1.23** |
| AUC | Wikipedia | $94.33_{\pm0.27}$ | $91.49_{\pm0.45}$ | $95.90_{\pm0.09}$ | $97.72_{\pm0.03}$ | $98.03_{\pm0.04}$ | $95.57_{\pm0.20}$ | $96.30_{\pm0.04}$ | $95.82_{\pm0.18}$ | $97.76_{\pm0.05}$ | $\underline{98.48_{\pm0.03}}$ | $\mathbf{98.90_{\pm0.01}}$ |
| | Reddit | $96.52_{\pm0.13}$ | $96.05_{\pm0.12}$ | $96.98_{\pm0.04}$ | $97.39_{\pm0.07}$ | $98.42_{\pm0.02}$ | $93.80_{\pm0.07}$ | $94.97_{\pm0.05}$ | $98.00_{\pm0.04}$ | $98.38_{\pm0.07}$ | $\underline{98.71_{\pm0.01}}$ | $\mathbf{98.73_{\pm0.02}}$ |
| | MOOC | $83.16_{\pm1.30}$ | $84.03_{\pm0.49}$ | $86.84_{\pm0.17}$ | $\underline{91.24_{\pm0.99}}$ | $81.86_{\pm0.25}$ | $81.43_{\pm0.19}$ | $82.77_{\pm0.24}$ | $84.72_{\pm1.31}$ | $90.27_{\pm0.96}$ | $87.62_{\pm0.51}$ | $\mathbf{95.55_{\pm0.25}}$ |
| | LastFM | $81.13_{\pm3.39}$ | $82.24_{\pm1.51}$ | $76.99_{\pm0.29}$ | $82.61_{\pm3.15}$ | $87.82_{\pm0.12}$ | $70.84_{\pm0.85}$ | $80.37_{\pm0.18}$ | $91.60_{\pm1.31}$ | $92.15_{\pm0.68}$ | $\underline{94.08_{\pm0.08}}$ | $\mathbf{95.36_{\pm0.06}}$ |
| | Enron | $81.96_{\pm1.34}$ | $76.34_{\pm4.20}$ | $64.63_{\pm1.74}$ | $78.83_{\pm1.11}$ | $87.02_{\pm0.50}$ | $72.33_{\pm0.99}$ | $76.51_{\pm0.71}$ | $87.95_{\pm0.58}$ | $\mathbf{90.69_{\pm0.61}}$ | $95.29_{\pm0.03}$ | $\underline{90.21_{\pm0.49}}$ |
| | Social Evo. | $93.70_{\pm0.29}$ | $91.18_{\pm0.49}$ | $93.41_{\pm0.19}$ | $93.43_{\pm0.59}$ | $84.73_{\pm0.27}$ | $93.71_{\pm0.18}$ | $94.09_{\pm0.07}$ | $88.15_{\pm6.36}$ | $94.78_{\pm0.37}$ | $\underline{95.29_{\pm0.03}}$ | $\mathbf{95.47_{\pm0.04}}$ |
| | UCI | $78.80_{\pm0.94}$ | $58.08_{\pm1.81}$ | $77.64_{\pm0.38}$ | $86.68_{\pm2.29}$ | $90.40_{\pm0.11}$ | $84.49_{\pm1.82}$ | $89.30_{\pm0.57}$ | $83.78_{\pm0.37}$ | $\underline{93.18_{\pm0.17}}$ | $92.63_{\pm0.13}$ | $\mathbf{94.40_{\pm0.03}}$ |
| | Flights | $95.21_{\pm0.32}$ | $93.56_{\pm0.70}$ | $88.64_{\pm0.35}$ | $95.92_{\pm0.43}$ | $96.86_{\pm0.02}$ | $82.48_{\pm0.01}$ | $82.27_{\pm0.06}$ | $96.97_{\pm0.20}$ | $97.69_{\pm0.05}$ | $\underline{97.80_{\pm0.02}}$ | $\mathbf{98.05_{\pm0.04}}$ |
| | Can. Parl. | $53.81_{\pm1.14}$ | $55.27_{\pm0.49}$ | $56.51_{\pm0.75}$ | $55.86_{\pm0.75}$ | $58.83_{\pm1.13}$ | $55.83_{\pm1.07}$ | $58.32_{\pm1.08}$ | $62.70_{\pm2.91}$ | $49.64_{\pm0.69}$ | $\mathbf{89.33_{\pm0.48}}$ | $\underline{69.21_{\pm1.31}}$ |
| | US Legis. | $58.12_{\pm2.35}$ | $61.07_{\pm0.56}$ | $48.27_{\pm3.50}$ | $62.38_{\pm0.48}$ | $51.49_{\pm1.13}$ | $50.43_{\pm1.48}$ | $47.20_{\pm0.89}$ | $\underline{64.22_{\pm0.65}}$ | $61.89_{\pm1.52}$ | $53.21_{\pm3.04}$ | $\mathbf{65.29_{\pm0.61}}$ |
| | UN Trade | $62.28_{\pm0.50}$ | $58.82_{\pm0.98}$ | $62.72_{\pm0.12}$ | $59.99_{\pm3.50}$ | $67.05_{\pm0.21}$ | $63.76_{\pm0.07}$ | $63.48_{\pm0.37}$ | $\underline{69.15_{\pm2.33}}$ | $64.05_{\pm0.72}$ | $67.25_{\pm1.05}$ | $\mathbf{86.88_{\pm0.23}}$ |
| | UN Vote | $58.13_{\pm1.43}$ | $55.13_{\pm3.46}$ | $51.83_{\pm1.35}$ | $\underline{61.23_{\pm2.71}}$ | $48.34_{\pm0.76}$ | $50.51_{\pm1.05}$ | $50.04_{\pm0.86}$ | $\mathbf{68.55_{\pm0.90}}$ | $59.01_{\pm1.88}$ | $56.73_{\pm0.69}$ | $54.82_{\pm4.04}$ |
| | Contact | $95.37_{\pm0.92}$ | $91.89_{\pm0.38}$ | $96.53_{\pm0.10}$ | $94.84_{\pm0.75}$ | $89.07_{\pm0.34}$ | $93.05_{\pm0.09}$ | $92.83_{\pm0.05}$ | $95.70_{\pm0.06}$ | $97.66_{\pm0.12}$ | $\underline{98.30_{\pm0.02}}$ | $\mathbf{98.51_{\pm0.01}}$ |
| | Avg. Rank | 7.46 | 8.62 | 7.92 | 5.69 | 6.46 | 8.77 | 8.00 | 4.77 | 3.92 | $\underline{2.77}$ | **1.62** |

## Table 7: Transductive results for historical negative sampling.

| Metric | Dataset | JODIE | DyRep | TGAT | TGN | CAWN | EdgeBank | TCL | GraphMixer | NAT | PINT | DyGFormer | TPNet |
|---|---|---|---|---|---|---|---|---|---|---|---|---|---|
| AP | Wikipedia | $83.01_{\pm0.66}$ | $79.93_{\pm0.56}$ | $87.38_{\pm0.22}$ | $86.86_{\pm0.33}$ | $71.21_{\pm1.67}$ | $73.35_{\pm0.00}$ | $89.05_{\pm0.39}$ | $90.90_{\pm0.10}$ | $68.36_{\pm4.47}$ | **$91.33_{\pm0.76}$** | $82.23_{\pm2.54}$ | $81.55_{\pm4.10}$ |
| | Reddit | $80.03_{\pm0.36}$ | $79.83_{\pm0.31}$ | $79.55_{\pm0.20}$ | $81.22_{\pm0.61}$ | $80.82_{\pm0.45}$ | $73.59_{\pm0.00}$ | $77.14_{\pm0.16}$ | $78.44_{\pm0.18}$ | $80.58_{\pm0.91}$ | **$83.93_{\pm1.11}$** | $81.57_{\pm0.67}$ | $81.02_{\pm1.31}$ |
| | MOOC | $78.94_{\pm1.25}$ | $75.60_{\pm1.12}$ | $82.19_{\pm0.62}$ | $87.06_{\pm1.93}$ | $74.05_{\pm0.95}$ | $60.71_{\pm0.00}$ | $77.06_{\pm0.41}$ | $77.77_{\pm0.92}$ | $85.42_{\pm3.26}$ | $84.80_{\pm1.57}$ | $85.85_{\pm0.66}$ | **$92.69_{\pm0.95}$** |
| | LastFM | $74.35_{\pm3.81}$ | $74.92_{\pm2.46}$ | $71.59_{\pm0.24}$ | $76.87_{\pm4.64}$ | $69.86_{\pm0.43}$ | $73.03_{\pm0.00}$ | $59.30_{\pm2.21}$ | $72.47_{\pm0.49}$ | $78.75_{\pm1.63}$ | $84.95_{\pm0.43}$ | $81.57_{\pm0.48}$ | **$87.74_{\pm0.50}$** |
| | Enron | $69.85_{\pm2.70}$ | $71.19_{\pm2.76}$ | $64.07_{\pm1.05}$ | $73.91_{\pm1.76}$ | $64.73_{\pm0.36}$ | $76.53_{\pm0.00}$ | $70.66_{\pm0.39}$ | $77.98_{\pm0.92}$ | $72.90_{\pm1.38}$ | **$85.41_{\pm0.84}$** | $75.63_{\pm0.73}$ | $80.79_{\pm1.68}$ |
| | Social Evo. | $87.44_{\pm6.78}$ | $93.29_{\pm0.43}$ | $95.01_{\pm0.44}$ | $94.45_{\pm0.56}$ | $85.53_{\pm0.38}$ | $80.57_{\pm0.00}$ | $94.74_{\pm0.31}$ | $94.93_{\pm0.31}$ | $91.29_{\pm4.89}$ | $96.73_{\pm0.29}$ | **$97.38_{\pm0.14}$** | $96.80_{\pm0.41}$ |
| | UCI | $75.24_{\pm5.80}$ | $55.10_{\pm3.14}$ | $68.27_{\pm1.37}$ | $80.43_{\pm2.12}$ | $65.30_{\pm0.45}$ | $65.50_{\pm0.00}$ | $80.25_{\pm2.74}$ | $84.11_{\pm1.35}$ | $75.38_{\pm1.28}$ | **$93.96_{\pm0.20}$** | $82.17_{\pm0.82}$ | $86.34_{\pm0.80}$ |
| | Flights | $66.48_{\pm2.59}$ | $67.61_{\pm0.99}$ | **$72.38_{\pm0.18}$** | $66.70_{\pm1.64}$ | $64.72_{\pm0.97}$ | $70.53_{\pm0.00}$ | $70.68_{\pm0.24}$ | $71.47_{\pm0.26}$ | $64.74_{\pm1.83}$ | $66.82_{\pm1.44}$ | $66.59_{\pm0.49}$ | $69.10_{\pm1.27}$ |
| | Can. Parl. | $51.79_{\pm0.63}$ | $63.31_{\pm1.23}$ | $67.13_{\pm0.84}$ | $68.42_{\pm3.07}$ | $66.53_{\pm2.77}$ | $63.84_{\pm0.00}$ | $65.93_{\pm3.00}$ | $74.34_{\pm0.87}$ | $77.72_{\pm1.78}$ | $63.84_{\pm5.36}$ | **$97.00_{\pm0.31}$** | $86.61_{\pm3.57}$ |
| | US Legis. | $51.71_{\pm5.76}$ | $86.88_{\pm2.25}$ | $62.14_{\pm6.60}$ | $74.00_{\pm7.57}$ | $68.82_{\pm8.23}$ | $63.22_{\pm0.00}$ | $80.53_{\pm3.95}$ | $81.65_{\pm1.02}$ | $91.12_{\pm1.97}$ | $73.58_{\pm4.16}$ | $85.30_{\pm3.88}$ | **$94.55_{\pm0.62}$** |
| | UN Trade | $61.39_{\pm1.83}$ | $59.19_{\pm1.07}$ | $55.74_{\pm0.91}$ | $58.44_{\pm5.51}$ | $55.71_{\pm0.38}$ | $81.32_{\pm0.00}$ | $55.90_{\pm1.17}$ | $57.05_{\pm1.22}$ | $78.65_{\pm1.16}$ | $70.07_{\pm2.28}$ | $64.41_{\pm1.40}$ | **$85.22_{\pm1.22}$** |
| | UN Vote | $70.02_{\pm0.81}$ | $69.30_{\pm1.12}$ | $52.96_{\pm2.14}$ | $69.37_{\pm3.93}$ | $51.26_{\pm0.04}$ | **$84.89_{\pm0.00}$** | $52.30_{\pm2.35}$ | $51.20_{\pm1.60}$ | $71.39_{\pm2.68}$ | $71.79_{\pm2.53}$ | $60.84_{\pm1.58}$ | $74.68_{\pm1.38}$ |
| | Contact | $95.31_{\pm2.13}$ | $96.39_{\pm0.20}$ | $96.05_{\pm0.52}$ | $93.05_{\pm2.35}$ | $84.16_{\pm0.49}$ | $88.81_{\pm0.00}$ | $93.86_{\pm0.21}$ | $93.36_{\pm0.41}$ | $96.84_{\pm0.57}$ | $97.61_{\pm0.18}$ | $97.57_{\pm0.06}$ | **$98.02_{\pm0.15}$** |
| | Avg. Rank | 8.23 | 7.69 | 7.54 | 5.92 | 10.31 | 8.08 | 7.77 | 6.23 | 5.85 | 3.62 | 4.23 | **2.46** |
| AUC | Wikipedia | $80.77_{\pm0.73}$ | $77.74_{\pm0.33}$ | $82.87_{\pm0.22}$ | $82.74_{\pm0.32}$ | $67.84_{\pm0.64}$ | $77.27_{\pm0.00}$ | $85.76_{\pm0.46}$ | $87.68_{\pm0.17}$ | $69.32_{\pm2.78}$ | **$89.25_{\pm0.49}$** | $78.80_{\pm1.95}$ | $79.89_{\pm2.47}$ |
| | Reddit | $80.52_{\pm0.32}$ | $80.15_{\pm0.18}$ | $79.33_{\pm0.16}$ | $81.11_{\pm0.19}$ | $80.27_{\pm0.30}$ | $78.58_{\pm0.00}$ | $76.49_{\pm0.16}$ | $77.80_{\pm0.12}$ | $79.36_{\pm0.83}$ | **$82.98_{\pm0.63}$** | $80.54_{\pm0.29}$ | $81.87_{\pm0.49}$ |
| | MOOC | $82.75_{\pm0.83}$ | $81.06_{\pm0.94}$ | $80.81_{\pm0.67}$ | $88.00_{\pm1.80}$ | $71.57_{\pm1.07}$ | $61.90_{\pm0.00}$ | $72.09_{\pm0.56}$ | $76.68_{\pm1.40}$ | $84.93_{\pm2.89}$ | $87.44_{\pm1.74}$ | $87.04_{\pm0.35}$ | **$93.45_{\pm0.67}$** |
| | LastFM | $75.22_{\pm2.36}$ | $74.65_{\pm1.98}$ | $64.27_{\pm0.26}$ | $77.97_{\pm3.04}$ | $67.88_{\pm0.24}$ | $78.09_{\pm0.00}$ | $47.24_{\pm3.13}$ | $64.21_{\pm0.73}$ | $75.89_{\pm2.21}$ | $81.89_{\pm0.96}$ | $78.78_{\pm0.35}$ | **$84.64_{\pm0.45}$** |
| | Enron | $75.39_{\pm2.37}$ | $74.69_{\pm3.55}$ | $61.85_{\pm1.43}$ | $77.09_{\pm2.22}$ | $65.10_{\pm0.34}$ | $79.59_{\pm0.00}$ | $67.95_{\pm0.88}$ | $75.27_{\pm1.14}$ | $73.22_{\pm2.18}$ | **$83.80_{\pm0.68}$** | $76.55_{\pm0.52}$ | $81.16_{\pm1.28}$ |
| | Social Evo. | $90.06_{\pm3.15}$ | $93.12_{\pm0.34}$ | $93.08_{\pm0.59}$ | $94.71_{\pm0.53}$ | $87.43_{\pm0.15}$ | $85.81_{\pm0.00}$ | $93.44_{\pm0.68}$ | $94.39_{\pm0.31}$ | $91.87_{\pm4.52}$ | $96.82_{\pm0.24}$ | **$97.28_{\pm0.07}$** | $97.22_{\pm0.30}$ |
| | UCI | $78.64_{\pm3.50}$ | $57.91_{\pm3.12}$ | $58.89_{\pm1.57}$ | $77.25_{\pm2.68}$ | $57.86_{\pm0.15}$ | $69.56_{\pm0.00}$ | $72.25_{\pm3.46}$ | $77.54_{\pm2.02}$ | $71.52_{\pm1.61}$ | **$92.05_{\pm0.36}$** | $76.97_{\pm0.24}$ | $80.42_{\pm0.64}$ |
| | Flights | $68.97_{\pm1.87}$ | $69.43_{\pm0.90}$ | $72.20_{\pm0.16}$ | $68.39_{\pm0.95}$ | $66.11_{\pm0.71}$ | **$74.64_{\pm0.00}$** | $70.57_{\pm0.18}$ | $70.37_{\pm0.23}$ | $67.11_{\pm2.16}$ | $69.52_{\pm0.90}$ | $68.09_{\pm0.43}$ | $71.82_{\pm0.82}$ |
| | Can. Parl. | $62.44_{\pm1.11}$ | $70.16_{\pm1.70}$ | $70.86_{\pm0.94}$ | $73.23_{\pm3.08}$ | $72.06_{\pm3.94}$ | $63.04_{\pm0.00}$ | $69.95_{\pm3.70}$ | $79.03_{\pm1.01}$ | $81.47_{\pm2.58}$ | $73.89_{\pm4.10}$ | **$97.61_{\pm0.40}$** | $86.39_{\pm3.73}$ |
| | US Legis. | $67.47_{\pm6.40}$ | $91.44_{\pm1.18}$ | $73.47_{\pm5.25}$ | $83.53_{\pm4.53}$ | $78.62_{\pm7.46}$ | $67.41_{\pm0.00}$ | $83.97_{\pm3.71}$ | $85.17_{\pm0.70}$ | $94.63_{\pm1.16}$ | $83.45_{\pm2.85}$ | $90.77_{\pm1.96}$ | **$96.28_{\pm0.44}$** |
| | UN Trade | $68.92_{\pm1.40}$ | $64.36_{\pm1.40}$ | $60.37_{\pm0.68}$ | $63.93_{\pm5.41}$ | $63.09_{\pm0.74}$ | $86.61_{\pm0.00}$ | $61.43_{\pm1.04}$ | $63.20_{\pm1.54}$ | $80.68_{\pm0.98}$ | $76.95_{\pm1.92}$ | $73.86_{\pm1.13}$ | **$88.90_{\pm1.00}$** |
| | UN Vote | $76.84_{\pm1.01}$ | $74.72_{\pm1.43}$ | $53.95_{\pm3.15}$ | $73.40_{\pm5.20}$ | $51.27_{\pm0.33}$ | **$89.62_{\pm0.00}$** | $52.29_{\pm2.39}$ | $52.61_{\pm1.44}$ | $76.47_{\pm2.15}$ | $76.61_{\pm3.05}$ | $64.27_{\pm1.78}$ | $78.43_{\pm1.09}$ |
| | Contact | $96.35_{\pm0.92}$ | $96.00_{\pm0.23}$ | $95.39_{\pm0.43}$ | $93.76_{\pm1.29}$ | $83.06_{\pm0.32}$ | $92.17_{\pm0.00}$ | $93.34_{\pm0.19}$ | $93.14_{\pm0.34}$ | $96.79_{\pm0.50}$ | $97.34_{\pm0.16}$ | $97.17_{\pm0.05}$ | **$97.73_{\pm0.11}$** |
| | Avg. Rank | 6.77 | 7.31 | 8.38 | 5.62 | 10.31 | 7.54 | 8.54 | 7.08 | 6.46 | 3.15 | 4.77 | **2.08** |

## Table 8: Inductive results for historical negative sampling.

| Metrics | Datasets | JODIE | DyRep | TGAT | TGN | CAWN | TCL | GraphMixer | NAT | PINT | DyGFormer | RPNet |
|---|---|---|---|---|---|---|---|---|---|---|---|---|
| AP | Wikipedia | $68.69_{\pm0.39}$ | $62.18_{\pm1.27}$ | $84.17_{\pm0.22}$ | $81.76_{\pm0.32}$ | $67.27_{\pm1.63}$ | $82.20_{\pm2.18}$ | **$87.60_{\pm0.30}$** | $47.37_{\pm4.39}$ | $78.22_{\pm2.90}$ | $71.42_{\pm4.43}$ | $71.28_{\pm4.33}$ |
| | Reddit | $62.34_{\pm0.54}$ | $61.60_{\pm0.72}$ | $63.47_{\pm0.36}$ | $64.85_{\pm0.85}$ | $63.67_{\pm0.41}$ | $60.83_{\pm0.25}$ | $64.50_{\pm0.26}$ | $61.51_{\pm0.73}$ | **$67.56_{\pm0.83}$** | $65.37_{\pm0.60}$ | $62.15_{\pm1.72}$ |
| | MOOC | $63.22_{\pm1.55}$ | $62.93_{\pm1.24}$ | $76.73_{\pm0.97}$ | $77.00_{\pm3.41}$ | $74.68_{\pm0.68}$ | $74.27_{\pm0.53}$ | $74.00_{\pm0.97}$ | $69.30_{\pm0.95}$ | $73.74_{\pm1.66}$ | $80.82_{\pm0.30}$ | **$81.85_{\pm1.60}$** |
| | LastFM | $70.39_{\pm4.31}$ | $71.45_{\pm1.76}$ | $76.27_{\pm0.25}$ | $66.65_{\pm6.11}$ | $71.33_{\pm0.47}$ | $65.78_{\pm0.65}$ | $76.42_{\pm0.24}$ | $70.21_{\pm0.78}$ | $77.96_{\pm1.55}$ | $76.35_{\pm0.52}$ | **$82.27_{\pm1.22}$** |
| | Enron | $65.86_{\pm3.71}$ | $62.08_{\pm2.27}$ | $61.40_{\pm1.31}$ | $62.91_{\pm1.16}$ | $60.70_{\pm0.36}$ | $67.11_{\pm0.62}$ | $72.37_{\pm1.37}$ | $62.69_{\pm1.02}$ | $80.47_{\pm1.52}$ | $67.07_{\pm0.62}$ | $74.60_{\pm1.35}$ |
| | Social Evo. | $88.51_{\pm0.87}$ | $88.72_{\pm1.10}$ | $93.97_{\pm0.54}$ | $90.66_{\pm1.62}$ | $79.83_{\pm0.38}$ | $94.10_{\pm0.31}$ | $94.01_{\pm0.47}$ | $84.89_{\pm4.62}$ | $95.75_{\pm1.06}$ | **$96.82_{\pm0.16}$** | $96.38_{\pm0.18}$ |
| | UCI | $63.11_{\pm2.27}$ | $52.47_{\pm2.06}$ | $70.52_{\pm0.93}$ | $70.78_{\pm2.06}$ | $65.30_{\pm0.45}$ | $76.71_{\pm1.00}$ | $81.66_{\pm0.49}$ | $51.56_{\pm0.95}$ | **$85.49_{\pm0.22}$** | $72.13_{\pm1.87}$ | $78.48_{\pm1.18}$ |
| | Flights | $61.01_{\pm1.65}$ | $62.83_{\pm1.31}$ | $64.72_{\pm0.36}$ | $59.31_{\pm1.43}$ | $56.82_{\pm0.57}$ | $64.50_{\pm0.25}$ | **$65.28_{\pm0.24}$** | $51.63_{\pm1.10}$ | $53.46_{\pm0.61}$ | $57.11_{\pm0.21}$ | $54.67_{\pm0.76}$ |
| | Can. Parl. | $52.60_{\pm0.88}$ | $52.28_{\pm0.31}$ | $56.72_{\pm0.47}$ | $54.42_{\pm0.77}$ | $57.14_{\pm0.07}$ | $55.71_{\pm0.74}$ | $55.84_{\pm0.73}$ | $61.56_{\pm2.68}$ | $50.61_{\pm1.76}$ | **$87.40_{\pm0.85}$** | $68.97_{\pm1.60}$ |
| | US Legis. | $52.94_{\pm2.11}$ | $62.10_{\pm1.41}$ | $51.83_{\pm3.95}$ | $61.18_{\pm1.10}$ | $55.56_{\pm1.71}$ | $53.87_{\pm1.41}$ | $52.03_{\pm1.02}$ | $64.61_{\pm3.02}$ | $59.37_{\pm1.84}$ | $56.31_{\pm3.46}$ | **$66.95_{\pm1.81}$** |
| | UN Trade | $55.46_{\pm1.19}$ | $55.49_{\pm0.84}$ | $54.59_{\pm0.52}$ | $52.80_{\pm3.19}$ | $55.00_{\pm0.36}$ | $54.94_{\pm0.29}$ | $70.04_{\pm2.07}$ | $57.35_{\pm1.65}$ | $53.20_{\pm1.07}$ | | **$78.83_{\pm0.53}$** |
| | UN Vote | $61.04_{\pm1.30}$ | $60.22_{\pm1.78}$ | $53.05_{\pm3.10}$ | $63.74_{\pm3.00}$ | $47.98_{\pm0.84}$ | $54.19_{\pm2.17}$ | $48.09_{\pm0.43}$ | $64.91_{\pm1.58}$ | **$65.75_{\pm2.86}$** | $52.63_{\pm1.26}$ | $65.24_{\pm1.62}$ |
| | Contact | $90.42_{\pm2.34}$ | $89.22_{\pm0.66}$ | **$94.15_{\pm0.45}$** | $88.13_{\pm1.50}$ | $74.20_{\pm0.80}$ | $90.44_{\pm0.17}$ | $89.91_{\pm0.36}$ | $84.13_{\pm1.78}$ | $90.68_{\pm0.46}$ | $93.56_{\pm0.52}$ | $93.56_{\pm0.57}$ |
| | Avg. Rank | 7.46 | 7.62 | 5.69 | 6.31 | 8.08 | 5.92 | 5.23 | 7.62 | 4.23 | 4.62 | **3.15** |
| AUC | Wikipedia | $61.86_{\pm0.53}$ | $57.54_{\pm1.09}$ | $78.38_{\pm0.20}$ | $75.75_{\pm0.29}$ | $62.04_{\pm0.65}$ | $79.79_{\pm0.96}$ | **$82.87_{\pm0.21}$** | $40.95_{\pm4.99}$ | $73.29_{\pm2.33}$ | $68.33_{\pm2.82}$ | $67.95_{\pm2.77}$ |
| | Reddit | $61.69_{\pm0.39}$ | $60.45_{\pm0.37}$ | $64.43_{\pm0.27}$ | $64.55_{\pm0.50}$ | **$64.94_{\pm0.21}$** | $61.43_{\pm0.26}$ | $64.27_{\pm0.13}$ | $58.32_{\pm0.63}$ | $64.02_{\pm0.46}$ | $64.81_{\pm0.25}$ | $62.37_{\pm0.83}$ |
| | MOOC | $64.48_{\pm1.64}$ | $64.23_{\pm1.29}$ | $74.08_{\pm0.27}$ | $77.69_{\pm3.55}$ | $71.68_{\pm0.94}$ | $69.82_{\pm0.32}$ | $72.53_{\pm0.84}$ | $67.99_{\pm1.68}$ | $75.92_{\pm2.48}$ | $80.77_{\pm0.63}$ | **$84.46_{\pm0.88}$** |
| | LastFM | $68.44_{\pm3.26}$ | $68.79_{\pm1.08}$ | $69.89_{\pm5.62}$ | $66.99_{\pm5.62}$ | $67.69_{\pm0.24}$ | $55.88_{\pm1.85}$ | $70.07_{\pm0.20}$ | $64.18_{\pm0.65}$ | $75.02_{\pm0.66}$ | $70.73_{\pm0.37}$ | **$77.10_{\pm0.78}$** |
| | Enron | $65.32_{\pm3.57}$ | $61.50_{\pm2.50}$ | $57.84_{\pm2.18}$ | $62.68_{\pm1.09}$ | $62.25_{\pm0.40}$ | $64.06_{\pm1.02}$ | $68.20_{\pm1.62}$ | $61.69_{\pm0.68}$ | $78.90_{\pm1.29}$ | $65.78_{\pm0.42}$ | $74.50_{\pm1.02}$ |
| | Social Evo. | $88.53_{\pm0.55}$ | $87.93_{\pm1.05}$ | $91.87_{\pm0.72}$ | $92.10_{\pm1.22}$ | $83.54_{\pm0.24}$ | $93.28_{\pm0.60}$ | $93.62_{\pm0.35}$ | $84.89_{\pm4.72}$ | $96.29_{\pm0.56}$ | **$96.91_{\pm0.09}$** | $96.76_{\pm0.14}$ |
| | UCI | $60.24_{\pm1.94}$ | $51.25_{\pm2.37}$ | $62.32_{\pm1.18}$ | $62.69_{\pm0.90}$ | $56.39_{\pm0.10}$ | $70.46_{\pm1.94}$ | $75.98_{\pm0.84}$ | $42.59_{\pm0.96}$ | **$81.34_{\pm0.29}$** | $65.55_{\pm1.01}$ | $71.35_{\pm0.84}$ |
| | Flights | $60.72_{\pm1.29}$ | $61.99_{\pm1.39}$ | $63.38_{\pm0.26}$ | $59.66_{\pm1.04}$ | $56.58_{\pm0.44}$ | **$63.48_{\pm0.23}$** | $63.30_{\pm0.19}$ | $48.39_{\pm1.37}$ | $49.76_{\pm0.89}$ | $56.05_{\pm0.21}$ | $53.08_{\pm0.89}$ |
| | Can. Parl. | $51.62_{\pm1.00}$ | $52.38_{\pm0.46}$ | $58.30_{\pm0.61}$ | $55.64_{\pm0.54}$ | $60.11_{\pm0.48}$ | $57.30_{\pm1.03}$ | $56.68_{\pm1.20}$ | $61.72_{\pm2.76}$ | $48.93_{\pm2.35}$ | **$88.68_{\pm0.74}$** | $69.11_{\pm1.18}$ |
| | US Legis. | $58.12_{\pm2.94}$ | $67.94_{\pm0.98}$ | $49.99_{\pm4.88}$ | $64.87_{\pm1.65}$ | $54.41_{\pm1.31}$ | $52.12_{\pm2.13}$ | $49.28_{\pm0.86}$ | $66.95_{\pm4.17}$ | $62.82_{\pm2.18}$ | $56.57_{\pm3.22}$ | **$68.37_{\pm1.62}$** |
| | UN Trade | $58.73_{\pm1.19}$ | $57.90_{\pm1.33}$ | $59.74_{\pm0.59}$ | $55.61_{\pm3.54}$ | $60.95_{\pm0.80}$ | $61.12_{\pm0.97}$ | $59.88_{\pm1.17}$ | $70.43_{\pm2.07}$ | $62.61_{\pm1.70}$ | $58.46_{\pm1.65}$ | **$80.70_{\pm1.03}$** |
| | UN Vote | $65.16_{\pm1.28}$ | $63.98_{\pm2.12}$ | $51.73_{\pm4.12}$ | $68.59_{\pm3.11}$ | $48.01_{\pm1.77}$ | $54.66_{\pm2.11}$ | $45.49_{\pm0.42}$ | $67.69_{\pm1.92}$ | **$69.47_{\pm3.01}$** | $53.85_{\pm2.02}$ | $65.75_{\pm1.85}$ |
| | Contact | $90.80_{\pm1.18}$ | $88.88_{\pm0.68}$ | $93.76_{\pm0.41}$ | $88.84_{\pm1.39}$ | $74.79_{\pm0.37}$ | $90.37_{\pm0.16}$ | $90.04_{\pm0.29}$ | $84.74_{\pm1.44}$ | $91.99_{\pm0.28}$ | **$94.14_{\pm0.26}$** | $93.47_{\pm0.43}$ |
| | Avg. Rank | 7.23 | 8.08 | 5.92 | 6.00 | 7.46 | 6.00 | 5.38 | 7.92 | 4.31 | 4.38 | **3.31** |

Table 9: Transductive results for inductive negative sampling.

| Metric | Dataset | JODIE | DyRep | TGAT | TGN | CAWN | EdgeBank | TCL | GraphMixer | NAT | PINT | DyGFormer | TPNet |
|---|---|---|---|---|---|---|---|---|---|---|---|---|---|
| AP | Wikipedia | 75.65±0.79 | 70.21±1.58 | 87.00±0.16 | 85.62±0.44 | 74.06±2.62 | 80.63±0.00 | 86.76±0.72 | **88.59±0.17** | 53.85±6.25 | 82.39±2.70 | 78.29±5.38 | 79.35±5.52 |
| | Reddit | 86.98±0.16 | 86.30±0.26 | 89.59±0.24 | 88.10±0.24 | **91.67±0.24** | 85.48±0.00 | 87.45±0.29 | 85.26±0.11 | 81.71±1.03 | 87.59±0.65 | 91.11±0.40 | 88.19±0.33 |
| | MOOC | 65.23±2.19 | 61.66±0.95 | 75.95±0.64 | 77.50±2.91 | 73.51±0.94 | 49.43±0.00 | 74.65±0.54 | 74.27±0.92 | 77.28±1.94 | 75.70±1.43 | 81.24±0.69 | **88.18±0.97** |
| | LastFM | 62.67±4.49 | 64.41±2.70 | 71.13±0.17 | 65.95±5.98 | 67.48±0.77 | 75.49±0.00 | 58.21±0.00 | 68.12±0.33 | 64.02±1.48 | 72.24±0.92 | 73.97±0.50 | **77.99±1.30** |
| | Enron | 68.96±0.98 | 67.79±1.53 | 63.94±1.36 | 70.89±2.72 | 75.15±0.58 | 73.89±0.00 | 71.29±0.32 | 75.01±0.79 | 70.34±1.73 | **80.23±1.37** | 77.41±0.89 | 75.36±1.81 |
| | Social Evo. | 89.82±4.11 | 93.28±0.48 | 94.84±0.44 | 95.13±0.56 | 88.32±0.27 | 83.69±0.00 | 94.90±0.36 | 94.72±0.33 | 92.12±4.53 | 97.15±0.25 | **97.68±0.10** | 97.35±0.32 |
| | UCI | 65.99±1.40 | 54.79±1.76 | 68.67±0.84 | 70.94±0.71 | 64.61±0.48 | 57.43±0.00 | 76.01±1.11 | 80.10±0.51 | 57.55±0.81 | **83.61±0.23** | 72.25±1.71 | 77.26±1.57 |
| | Flights | 69.07±4.02 | 70.57±1.82 | 75.48±0.26 | 71.09±2.72 | 69.18±1.52 | **81.08±0.00** | 74.62±0.18 | 74.87±0.21 | 59.16±1.48 | 61.99±1.70 | 70.92±1.78 | 64.78±1.50 |
| | Can. Parl. | 48.42±0.66 | 58.61±0.86 | 68.82±1.21 | 65.34±2.87 | 67.75±1.00 | 62.16±0.00 | 65.85±1.75 | 69.48±0.63 | 78.03±1.27 | 59.15±6.20 | **95.44±0.57** | 85.59±3.08 |
| | US Legis. | 50.27±5.13 | 83.44±1.16 | 61.91±5.82 | 67.57±6.47 | 65.81±8.52 | 64.74±0.00 | 78.15±3.34 | 79.63±0.84 | 88.40±2.34 | 67.78±4.29 | 81.25±3.62 | **91.05±1.21** |
| | UN Trade | 60.42±1.48 | 60.19±1.24 | 60.61±1.24 | 61.04±6.01 | 62.54±0.67 | 72.97±0.00 | 61.06±1.74 | 60.15±1.29 | 78.38±2.24 | 69.72±1.84 | 55.79±1.02 | **86.61±0.99** |
| | UN Vote | 67.79±1.46 | 67.53±1.98 | 52.89±1.61 | 67.63±2.67 | 52.19±0.34 | 66.30±0.00 | 50.62±0.82 | 51.60±0.73 | 72.53±1.94 | 68.37±1.92 | 51.91±0.84 | **75.05±1.41** |
| | Contact | 93.43±1.78 | 94.18±0.10 | 94.35±0.48 | 90.18±3.28 | 89.31±0.27 | 85.20±0.00 | 91.35±0.21 | 90.87±0.35 | 91.13±1.48 | 94.05±0.52 | 94.75±0.28 | **95.84±0.32** |
| | Avg. Rank | 9.08 | 8.62 | 5.92 | 6.23 | 7.62 | 7.77 | 6.69 | 6.38 | 7.31 | 5.08 | 4.46 | **2.85** |
| AUC | Wikipedia | 70.96±0.78 | 67.36±0.96 | 81.93±0.22 | 80.97±0.31 | 70.95±0.95 | 81.73±0.00 | 82.19±0.48 | **84.28±0.30** | 52.11±5.83 | 77.03±1.93 | 75.09±3.70 | 75.36±3.41 |
| | Reddit | 83.51±0.15 | 82.90±0.31 | 87.13±0.20 | 84.56±0.24 | **88.04±0.29** | 85.93±0.00 | 84.67±0.29 | 82.21±0.13 | 76.01±1.04 | 81.02±0.68 | 86.23±0.51 | 81.64±0.42 |
| | MOOC | 66.63±2.30 | 63.26±1.01 | 73.18±0.33 | 77.44±2.86 | 70.32±1.43 | 48.18±0.00 | 70.36±0.37 | 72.45±0.72 | 75.56±2.39 | 77.00±2.24 | 80.76±0.76 | **89.07±0.63** |
| | LastFM | 61.32±3.49 | 62.15±2.12 | 63.99±0.21 | 65.46±4.27 | 67.92±0.44 | **77.37±0.00** | 46.93±2.59 | 60.22±0.32 | 58.54±2.37 | 69.51±0.61 | 69.25±0.36 | 72.48±0.90 |
| | Enron | 70.92±1.05 | 68.73±1.34 | 60.45±2.12 | 71.34±2.46 | 75.17±0.50 | 75.00±0.00 | 67.64±0.86 | 71.53±0.85 | 69.62±1.12 | **78.91±0.96** | 74.07±0.64 | 75.44±1.38 |
| | Social Evo. | 90.01±3.19 | 93.07±0.38 | 92.94±0.61 | 95.24±0.56 | 89.93±0.15 | 87.88±0.00 | 93.44±0.72 | 94.22±0.32 | 92.53±4.20 | 97.10±0.22 | 97.51±0.06 | **97.59±0.25** |
| | UCI | 64.14±1.26 | 54.25±2.01 | 60.80±1.01 | 64.11±1.04 | 58.06±0.26 | 58.03±0.00 | 70.05±1.86 | 74.59±0.74 | 49.73±1.20 | **79.42±0.31** | 65.96±1.18 | 70.85±0.96 |
| | Flights | 69.99±3.10 | 71.13±1.55 | 73.47±0.18 | 71.63±1.72 | 69.70±0.75 | **81.10±0.00** | 72.54±0.19 | 72.21±0.21 | 59.73±1.61 | 59.61±1.65 | 69.53±1.17 | 64.21±1.30 |
| | Can. Parl. | 52.88±0.80 | 63.53±0.65 | 72.47±1.18 | 69.57±2.81 | 72.93±1.78 | 61.41±0.00 | 69.47±2.12 | 70.52±0.94 | 80.03±1.75 | 65.30±6.98 | **96.70±0.59** | 85.05±2.71 |
| | US Legis. | 59.05±5.52 | 89.44±0.71 | 71.62±5.42 | 78.12±4.46 | 76.45±7.02 | 68.66±0.00 | 82.54±3.91 | 84.22±0.91 | 93.04±1.35 | 79.09±3.45 | 87.96±1.80 | **94.48±0.50** |
| | UN Trade | 66.82±1.27 | 65.60±1.28 | 66.13±0.78 | 66.37±5.39 | 71.73±0.74 | 74.20±0.00 | 67.80±1.21 | 66.53±1.22 | 80.77±1.31 | 75.76±1.54 | 62.56±1.51 | **89.56±0.87** |
| | UN Vote | 73.73±1.61 | 72.80±2.16 | 53.04±2.58 | 72.69±3.72 | 52.75±0.90 | 72.85±0.00 | 52.02±1.64 | 51.89±0.74 | 77.36±1.52 | 74.11±2.51 | 53.37±1.26 | **79.13±1.16** |
| | Contact | 94.47±1.08 | 94.23±0.18 | 94.10±0.41 | 91.64±1.72 | 87.68±0.24 | 85.87±0.00 | 91.23±0.19 | 90.96±0.27 | 92.51±1.16 | 94.71±0.33 | 95.01±0.15 | **95.93±0.23** |
| | Avg. Rank | 8.08 | 8.23 | 6.77 | 6.31 | 7.31 | 7.00 | 7.00 | 6.54 | 7.46 | 5.08 | 5.00 | **3.23** |

Table 10: Inductive results for inductive negative sampling.

| Metrics | Datasets | JODIE | DyRep | TGAT | TGN | CAWN | TCL | GraphMixer | NAT | PINT | DyGFormer | RPNet |
|---|---|---|---|---|---|---|---|---|---|---|---|---|
| AP | Wikipedia | 68.70±0.39 | 62.19±1.28 | 84.17±0.22 | 81.77±0.32 | 67.24±1.63 | 82.20±2.18 | **87.60±0.29** | 47.37±4.39 | 78.23±2.89 | 71.42±4.43 | 71.29±4.33 |
| | Reddit | 62.32±0.54 | 61.58±0.72 | 63.40±0.36 | 64.84±0.84 | 63.65±0.41 | 60.81±0.26 | 64.49±0.25 | 61.51±0.73 | **67.56±0.83** | 65.35±0.60 | 62.14±1.72 |
| | MOOC | 63.22±1.55 | 62.92±1.24 | 76.72±0.30 | 77.07±3.40 | 74.69±0.68 | 74.28±0.53 | 73.99±0.97 | 69.30±0.95 | 73.74±1.06 | 80.82±0.52 | **81.85±1.60** |
| | LastFM | 70.39±4.31 | 71.45±1.75 | 76.28±0.25 | 69.46±4.65 | 71.33±0.47 | 65.78±0.05 | 76.42±0.32 | 70.21±0.78 | 77.96±1.55 | 76.35±0.52 | **82.27±1.22** |
| | Enron | 65.86±3.71 | 62.08±2.27 | 61.40±1.30 | 62.90±1.16 | 60.72±0.36 | 67.11±0.62 | 72.37±1.38 | 62.69±1.02 | **80.47±1.52** | 67.07±0.62 | 74.60±1.35 |
| | Social Evo. | 88.51±0.87 | 88.72±1.10 | 93.97±0.54 | 90.65±1.62 | 79.83±0.39 | 94.10±0.32 | 94.01±0.47 | 84.89±4.62 | 95.75±1.06 | **96.82±0.17** | 96.38±0.18 |
| | UCI | 63.16±2.27 | 52.47±2.09 | 70.49±0.93 | 70.73±0.99 | 64.54±0.47 | 76.65±0.99 | 81.64±0.49 | 51.58±0.78 | **85.50±0.22** | 72.13±1.86 | 78.50±1.18 |
| | Flights | 61.01±1.66 | 62.83±1.31 | 64.72±0.37 | 59.32±1.45 | 56.82±0.56 | 64.50±0.25 | **65.29±0.24** | 51.60±1.11 | 53.43±0.61 | 57.11±0.20 | 54.63±0.75 |
| | Can. Parl. | 52.58±0.86 | 52.24±0.28 | 56.46±0.50 | 54.18±0.73 | 57.06±0.08 | 55.46±0.69 | 55.76±0.65 | 61.71±2.77 | 50.66±1.67 | **87.22±0.82** | 68.87±1.68 |
| | US Legis. | 52.94±2.11 | 62.10±1.41 | 51.83±3.95 | 61.18±1.10 | 55.56±1.71 | 53.87±1.41 | 52.03±1.02 | 64.61±3.02 | 59.37±1.84 | 56.31±3.46 | **66.95±1.81** |
| | UN Trade | 55.43±1.20 | 55.42±0.87 | 55.58±0.68 | 52.80±3.24 | 54.97±0.38 | 54.88±3.01 | 54.65±1.90 | 59.96±2.08 | 57.26±1.68 | 52.56±1.70 | **78.95±0.52** |
| | UN Vote | 61.17±1.33 | 60.29±1.79 | 53.08±3.10 | 63.71±2.97 | 48.01±0.82 | 54.13±2.16 | 48.10±0.40 | 64.81±1.54 | **65.70±2.82** | 52.61±1.25 | 65.33±1.59 |
| | Contact | 90.43±2.33 | 89.22±0.65 | **94.14±0.45** | 88.12±1.50 | 74.19±0.81 | 90.43±0.17 | 89.91±0.36 | 84.13±1.77 | 90.68±0.46 | 93.55±0.52 | 93.57±0.57 |
| | Avg. Rank | 7.38 | 7.77 | 5.54 | 6.23 | 8.08 | 5.92 | 5.23 | 7.62 | 4.23 | 4.77 | **3.15** |
| AUC | Wikipedia | 61.87±0.53 | 57.54±1.09 | 78.38±0.20 | 75.76±0.29 | 62.02±0.65 | 79.79±0.96 | **82.88±0.21** | 40.95±4.99 | 73.30±2.33 | 68.33±2.82 | 67.96±2.77 |
| | Reddit | 61.69±0.39 | 60.44±0.37 | 64.39±0.27 | 64.55±0.50 | **64.91±0.21** | 61.36±0.26 | 64.27±0.13 | 58.31±0.63 | 64.02±0.46 | 64.80±0.25 | 62.37±0.83 |
| | MOOC | 64.48±1.64 | 64.22±1.29 | 74.07±0.27 | 77.68±3.55 | 71.69±0.94 | 69.83±0.32 | 72.52±0.84 | 68.00±1.68 | 75.93±2.48 | 80.77±0.63 | **84.46±0.88** |
| | LastFM | 68.44±3.26 | 68.79±1.08 | 69.89±0.26 | 66.99±5.61 | 67.68±0.24 | 55.88±1.95 | 70.07±0.20 | 64.18±0.65 | 75.02±0.66 | 70.73±0.37 | **77.10±0.78** |
| | Enron | 65.32±3.57 | 61.50±2.50 | 57.83±2.18 | 62.68±1.09 | 62.27±0.40 | 64.05±1.02 | 68.19±1.63 | 61.69±0.68 | **78.90±1.29** | 65.79±0.42 | 74.50±1.02 |
| | Social Evo. | 88.53±0.55 | 87.93±1.05 | 91.88±0.72 | 92.10±1.22 | 83.54±0.24 | 93.28±0.60 | 93.62±0.35 | 84.89±4.72 | 96.29±0.56 | **96.91±0.09** | 96.76±0.14 |
| | UCI | 60.27±1.94 | 51.26±2.40 | 62.29±1.17 | 62.66±0.91 | 56.39±0.11 | 70.42±1.93 | 75.97±0.85 | 42.63±0.97 | **81.35±0.29** | 65.58±1.00 | 71.37±0.84 |
| | Flights | 60.72±1.29 | 61.99±1.39 | 63.40±0.26 | 59.66±1.05 | 56.58±0.44 | **63.49±0.23** | 63.32±0.19 | 48.36±1.37 | 49.74±0.90 | 56.05±0.12 | 53.05±0.24 |
| | Can. Parl. | 51.61±0.98 | 52.35±0.52 | 58.15±0.62 | 55.43±0.42 | 60.01±0.47 | 56.88±0.93 | 56.63±1.09 | 61.83±2.92 | 48.93±2.24 | **88.51±0.73** | 68.98±1.21 |
| | US Legis. | 58.12±2.94 | 67.94±0.98 | 49.99±4.88 | 64.87±1.65 | 54.41±1.31 | 52.12±2.13 | 49.28±0.86 | 66.95±4.17 | 62.82±2.18 | 56.57±3.22 | **68.37±1.62** |
| | UN Trade | 58.71±1.20 | 57.87±1.36 | 59.98±0.59 | 55.62±3.59 | 60.88±0.79 | 61.01±0.93 | 59.71±1.17 | 70.34±2.04 | 62.54±1.73 | 57.28±3.06 | **80.78±1.03** |
| | UN Vote | 65.29±1.30 | 64.10±2.10 | 51.78±4.14 | 68.58±3.08 | 48.04±1.76 | 54.65±2.20 | 45.57±0.41 | 67.61±1.90 | **69.44±2.98** | 53.87±2.01 | 65.85±1.84 |
| | Contact | 90.80±1.18 | 88.87±0.67 | 93.76±0.40 | 88.85±1.39 | 74.79±0.38 | 90.37±0.16 | 90.04±0.29 | 84.74±1.44 | 92.00±0.28 | **94.14±0.26** | 93.47±0.43 |
| | Avg. Rank | 7.23 | 8.00 | 5.85 | 6.00 | 7.46 | 6.00 | 5.46 | 7.92 | 4.31 | 4.46 | **3.31** |

