# OpenReview forum: "Improving Temporal Link Prediction via Temporal Walk Matrix Projection"
_NeurIPS.cc/2024/Conference — NeurIPS 2024 poster_

### Official Review · Reviewer_YNEN · 2024-07-10

**Soundness:** 3
**Presentation:** 3
**Contribution:** 3
**Rating:** 7
**Confidence:** 2

**Summary:**

The paper introduces a framework for analysis of relative encodings as a function of random walk matrices, and a new model for temporal link prediction. The new model offers SOTA performance on multiple link prediction datasets, and achieves this performance more efficiently than current best models. The design is of the model is explained in detail.

**Strengths:**

This is an overall good paper which proposes and evaluates a new SOTA approach for temporal link prediction. It also introduces an original framework for unifying a range of existing methods.

1. The provided code is sufficiently commented, and detailed instructions for setup are provided, along with utility functions to start the training process.
2. Extensive evaluation has been done of the proposed model, and SOTA performance across a range of datasets has been demonstrated.
3. Current methods have been posed in the newly proposed framework, which is an interesting analytical contribution.

Clearly presented and significant contribution to the field of temporal link prediction.

**Weaknesses:**

1. The limitations of the approach are only briefly discussed in the appendix.
2. Although the code is well presented, it did not run out of the box for me. Some effort was required to install additional dependencies not listed in the requirements.txt, and to fix a runtime error in the sampling algorithm of the DataLoader.

Minor points / text linting suggestions:

Line 81: What is a "unified formation", did you mean "formulation"?

Line 94: Style "all the interactions happen" -> "all the interactions that happen"

Line 128: Doesn't A^(i) aggregate the i-step temporal walks, instead of k-step? Typo?

Line 187: "The value of score function.." -> "The value of the score function".

Line 202: What is d? The node degree or dimensionality? Please define. Also in line 2, Algorithm 1.

Line 203: Typo enumerating the H vectors. The index 1 is repeated, whereas it should be followed by 2.

Line 205: Style "Pseudocode code" -> "Pseudocode"

Line 211: Having the integer "i" in the exponent of e could be misleading for a reader who is skimming to get an overview of the paper. This letter is usually reserved as the imaginary unit when exponentiating e, so ideally pick another letter as the index.

Line 261: "and obtained" -> "and obtain"

**Questions:**

1. Line 265: Why does ReLU reduce estimation error? Is this an empirical result, or an architectural consideration specific to your data? Please clarify.

2. Line 43: The second argument sounds a bit vague. Could specific examples be mentioned for methods that use and don't use temporal information in constructing their embeddings?

3. Minor point on code: distutils has been removed in python 3.12 and above. To run the code I've had to install the following dependencies in addition to the ones listed: setuptools, numba, sklearn. I've had to fix the following issue.

```{python}
set(random.sample(test_node_set, int(0.1 * num_total_unique_node_ids)))
```
Had to be changed to:
```{python}
set(random.sample(sorted(test_node_set), int(0.1 * num_total_unique_node_ids)))
```

This happened installing your code in a new python 3.12 environment.

**Limitations:**

The limitations are briefly discussed in the appendix.

---

> ### Author Rebuttal · Authors · 2024-08-07
>
> Thank you for your valuable comments. In response, we have clarified how ReLU can reduce estimation error and discussed the utilization of temporal information in existing methods. We have also revised the paper and code according to your suggestions. We hope this addresses your concerns and are happy to answer any further questions you may have.
>
> **Q1 Line 265: Why does ReLU reduce estimation error? Is this an empirical result, or an architectural consideration specific to your data? Please clarify.**
>
> It is an architectural consideration specific to our construction of temporal walk matrices. Recall that our score function $s(\\cdot)$ for a temporal walk $W=[(w\_0,t\_0),(w\_1,t\_1),\\cdots,(w\_k,t\_k)]$ is $s(W)=\\prod\_{i=1}^k e^{-\lambda(t-t\_i)}$, which is always larger than zero.  Therefore, each element of the corresponding temporal walks matrices $A\_{u,v}^{(k)}(t) = \\sum\_{W \\in \\mathcal{M}\_{u,~v}^k} s(W)$ should be non-negative. From theorem 2, we know that the inner product of the node representations is approximately equal to the inner product of rows from temporal walk matrices, which is $\\langle \\bar{\\boldsymbol H}\_u^{(i)}(t), \\bar{\\boldsymbol H}\_v^{(j)}(t) \\rangle \\approx \\langle \\boldsymbol A\_u^{(i)}(t),\\boldsymbol A\_v^{(j)}(t) \\rangle$ (appears in line 225).  In line 265, each element of the raw pairwise feature $ \\tilde{\\boldsymbol f}\_{u ,v}$ is the inner product of node representations from $u$ and $v$ , which can be considered as an estimation of the inner product between the u-th and v-th rows of the temporal walk matrices. Since each element of temporal walk matrices is non-negative, the inner product of two rows from temporal matrices should also be non-negative. Thus we feed the raw pairwise feature into RELU to reduce estimation error.  Specifically, consider the l-th element of raw pairwise feature $ \\tilde f\_{u,v}[l]$ and assume it is the inner product of $\\bar{\\boldsymbol H}\_u^{(i)}(t)$ and $\\bar{\\boldsymbol H}\_v^{(j)}(t)$, which can be considered as an estimation of the inner product of $\\boldsymbol A\_u^{(i)}(t)$ and $\\boldsymbol A\_v^{(j)}(t)$. Then if $\\tilde f\_{u,v}[l]$ < 0,  using $|\\cdot|$ to denote the absolute vlaue , we will have $\\left |\\langle \\boldsymbol A\_u^{(i)}(t),\\boldsymbol A\_v^{(j)}(t) \\rangle - \\tilde f\_{u,v}[l]\\right| >\\left |\\langle \\boldsymbol A\_u^{(i)}(t),\\boldsymbol A\_v^{(j)}(t) \\rangle - 0\\right |$ since $\\langle \\boldsymbol A\_u^{(i)}(t),\\boldsymbol A\_v^{(j)}(t) \\rangle \\geq 0$ . Therefore, setting the negative value to zero by feeding $\\tilde{\\boldsymbol f}\_{u ,v}$ into RELU will have a lower estimation error than using the original value.
>
> **Q2 Line 43: The second argument sounds a bit vague. Could specific examples be mentioned for methods that use and don't use temporal information in constructing their embeddings?**
>
> Sure. We list the score function $s(\cdot)$ of methods that are analyzed in Section 2.2.2 in the following table, where $W=[(w_0,t_0),..,(w_k,t_k)]$ indicates a temporal walk and $Z_i= \\sum\_{\\{(w',w),t'\\} \\in \\mathcal E\_{w\_{~i},~~t\_i}} \\exp(-\alpha (t\_i-t'))$ indicates the normalize term of CAWN.
>
> | DyGFormer | PINT     | NAT      | CAWN                                                         |
> | --------- | -------- | -------- | ------------------------------------------------------------ |
> | $s(W)=1$  | $s(W)=1$ | $s(W)=1$ | $s(W)=\\prod\_{i=0}^{k-1}\\frac{\\exp(-\alpha(t\_i-t\_{i+1}))}{Z_i}$ |
>
>
> As shown in the above table, only CAWN considers the temporal information carried by a temporal walk. The score functions of the other three methods always yield one, so the element of their temporal walk matrices merely counts the number of temporal walks, ignoring the temporal information. For more discussion about existing methods, please refer to the conclusion starting from line 163 of the paper.
>
> **Q3 & W2: Although the code is well presented, it did not run out of the box for me. Some effort was required to install additional dependencies not listed in the requirements.txt, and to fix a runtime error in the sampling algorithm of the DataLoader.**
>
> Sorry for the incomplete information regarding the required environment.  The Python version used in our experiments is Python 3.9. We will add this information to the `requirements.txt` file and review our dependencies to ensure that no necessary packages have been omitted. Thanks for your efforts to adapt our code to the latest environment and we will revise our code according to your suggestion.
>
> **W1 The limitations of the approach are only briefly discussed in the appendix.**
>
> We will move the limitation section into the main body of our paper and expand the discussion.
>
> **W3 Minor points / text linting suggestions**
>
> We have revised the paper according to your suggestions for lines 81, 94, 187, 203, 205, 211, and 261.
>
> For line 128: Yes, it is a typo, and we will correct it.
>
> For line 202: $d$ indicates the dimensionality, and we will replace it with $d_R$.

---

> > ### Comment · Reviewer_YNEN · 2024-08-08
> >
> > Thank you for your response and clarifications.
> >
> > As it stands, I have no more outstanding concerns, and am happy to give this work an Accept.

---

> > > ### Author Response · Authors · 2024-08-09
> > > **Response to Reviewer YNEN**
> > >
> > > Thanks again! We sincerely appreciate your timely reply and support for our work.

---

### Official Review · Reviewer_SMMz · 2024-07-11

**Soundness:** 3
**Presentation:** 3
**Contribution:** 3
**Rating:** 6
**Confidence:** 1

**Summary:**

Based on the analysis of traditional methods, this paper proposes a unified framework for relative encoding and introduces a new temporal neural network, TPNet. This model not only addresses the high time complexity issues of traditional methods but also enhances relative encoding by incorporating factors such as time decay effects.

**Strengths:**

1、The authors propose a unified framework for relative encoding, treating them as a function of temporal random walk matrices.

2、The temporal random walk matrix not only takes into account temporal and structural information but also incorporates the effects of time decay.

3、To reduce computational complexity, the authors proposed an approximation scheme for the temporal random walk matrix, detailed in Algorithm 1.

4、The authors conducted thorough axiomatic proofs, providing solid theoretical support for the effectiveness of the TPNet network

**Weaknesses:**

1、If the authors could provide a comparison of memory usage, it would make the model more convincing.

2、This paper does not provide an analysis of the impact of different functions g on relative encoding.

3、The author did not compare with the latest models (published in AAAI 2024 or WWW 2024) in Table 1.

**Questions:**

please see weaknesses

---

> ### Author Rebuttal · Authors · 2024-08-07
>
> Thanks for your helpful reviews. In response, we have reported the memory usage of different methods, discussed the construction of our decoding function $g(\cdot)$, and compared TPNet with a new temporal link prediction method. We hope this addresses your concerns and are happy to answer any further questions you may have.
>
> **W1. If the authors could provide a comparison of memory usage, it would make the model more convincing.**
>
> We report the peak GPU memory usage in the following table, measured in MB. The last line of the table reports the relative memory usage,  calculated by dividing each method's memory usage by TCL's (i.e., method with the smallest average memory usage) and then averaging across datasets.
>
> |Dataset|TCL|JODIE|DyRep|TGN|TPNet (ours)|NAT|TGAT|GraphMixer|DyGFormer|CAWN|PINT|
> |:-----:|:-:|:---:|:---:|:-:|:---------:|:-:|:-:|:--------:|:-------:|:-:|:-:|
> |Wikipedia|171|247|253|271|299|430|570|714|279|633|2278|
> |Reddit|510|838|839|898|663|1265|909|1049|645|972|3381|
> |MOOC|336|446|459|464|470|835|735|877|673|1279|1405|
> |LastFM|911|1022|1038|1039|1044|1960|1309|1450|1643|2818|1113|
> |Enron|145|113|136|138|254|361|542|684|483|606|862|
> |Social Evo.|1439|1406|1430|1431|1504|2938|1838|1980|1547|2382|2154|
> |UCI|102|90|111|111|205|224|502|643|210|1045|890|
> |Flights|1335|2190|2192|2259|1503|2989|1733|1875|1673|2277|5532|
> |Can. Parl.|110|63|85|86|211|239|509|651|3150|2017|818|
> |US Legis.|101|62|85|86|201|297|500|642|440|563|810|
> |UN Trade|394|376|399|400|514|811|793|935|733|1337|1124|
> |UN Vote|741|735|759|760|870|1592|1140|1282|926|1684|774|
> |Contact|1653|1791|1813|1814|1794|3428|2052|2195|1761|2596|2544|
> |Rel. Memory|1.00|1.08|1.16|1.18|1.46|2.31|2.64|3.23|3.97|4.73|5.12|
>
> As shown in the above table, TPNet has lower relative memory usage than other link-wise methods (i.e., DyGFormer, NAT, CWAN, and PINT), showing its efficient memory usage. Additionally, on average, TPNet’s memory usage is 1.46 times that of the most memory-efficient method (i.e., TCL). Considering TPNet's SOTA performance and superior computational efficiency, this level of memory usage is satisfactory.
>
> **W2. This paper does not provide an analysis of the impact of different functions g on relative encoding.**
>
> The function $g(\cdot)$ introduced in Equation (3) can be considered a decoding function that extracts pairwise information from the constructed temporal walk matrices.  Unlike existing methods that use predefined $g(\cdot)$, we design it as a learnable function. Specifically, we feed the raw pairwise feature into an MLP ( line 265 of the paper), which serves as our decoding function $g(\cdot)$. By optimizing the learnable parameters in the MLP, we can adaptively determine the most suitable function $g(\cdot)$ for different graphs, better modeling graph dynamics.
>
> **W3. The author did not compare with the latest models (published in AAAI 2024 or WWW 2024) in Table 1**
>
> After reviewing the papers published in AAAI 2024 and WWW 2024, we did not find any that were closely related to our work. However, we found an ICLR 2024 paper that proposes a new temporal link prediction model called FreeDyG [1], which adopts a Fourier Transformer to enhance the node representation learning process. We report AP under random negative sampling for performance comparison.
>
> |                      |         |    Wikipedia     |      Reddit      |       MOOC       |      LastFM      |      Enron       |   Social Evo.    |       UCI        |
> | -------------------- | ------- | :--------------: | :--------------: | :--------------: | :--------------: | :--------------: | :--------------: | :--------------: |
> | Transductive Setting | FreeDyG |   99.26 ± 0.01   | **99.48 ± 0.01** |   89.61 ± 0.19   |   92.15 ± 0.16   |   92.51 ± 0.05   | **94.91 ± 0.01** |   96.28 ± 0.11   |
> |                      | TPNet   | **99.32 ± 0.03** |   99.27 ± 0.00   | **96.39 ± 0.09** | **94.50 ± 0.08** | **92.90 ± 0.17** |   94.73 ± 0.02   | **97.35 ± 0.04** |
> | Inductive Setting    | FreeDyG | **98.97 ± 0.01** | **98.91 ± 0.01** |   87.75 ± 0.62   |   94.89 ± 0.01   |   89.69 ± 0.17   | **94.76 ± 0.05** |   94.85 ± 0.10   |
> |                      | TPNet   |   98.91 ± 0.01   |   98.86 ± 0.01   | **95.07 ± 0.26** | **95.36 ± 0.11** | **90.34 ± 0.28** |   93.24 ± 0.07   | **95.74 ± 0.05** |
>
> As shown in the above table, TPNet significantly outperforms FreeDyG on MOOC, LastFM, and UCI, while exhibiting similar performance on other datasets, demonstrating its superiority.  We also report the inference time (s/epoch) to compare efficiency.
>
> |         |  Wikipedia   |    Reddit     |     MOOC      |    LastFM     |     Enron     |  Social Evo.  | UCI           |
> | :-----: | :----------: | :-----------: | :-----------: | :-----------: | :-----------: | :-----------: | ------------- |
> | FreeDyG |    12.37     |     61.11     |     32.19     |    102.73     |     9.75      |    163.94     | 4.75          |
> |  TPNet  |     2.51     |     11.10     |     6.38      |     22.71     |     1.95      |     33.34     | 0.93          |
> | Speedup | 4.93$\times$ | 5.51 $\times$ | 5.05 $\times$ | 4.52 $\times$ | 5.00 $\times$ | 4.92 $\times$ | 5.11 $\times$ |
>
> As shown in the above table, TPNet is more computationally efficient than FreeDyG, achieving at least a 4.52 $\times$ speedup.
>
> [1] Tian, Yuxing, Yiyan Qi, and Fan Guo. "FreeDyG: Frequency Enhanced Continuous-Time Dynamic Graph Model for Link Prediction." *The Twelfth International Conference on Learning Representations*. 2024.

---

> > ### Comment · Reviewer_SMMz · 2024-08-11
> > **Response**
> >
> > Thanks the authors for the rebuttal, and I have no more concerns.

---

> > > ### Author Response · Authors · 2024-08-12
> > > **Response to Reviewer SMMz**
> > >
> > > Thank you for your reply. We are pleased to hear that you have no further concerns.

---

### Official Review · Reviewer_vApy · 2024-07-11

**Soundness:** 3
**Presentation:** 3
**Contribution:** 3
**Rating:** 8
**Confidence:** 2

**Summary:**

The article investigates the application of relative encodings in the task of link prediction over temporal networks. Initially, the authors formally unify previously used relative encodings, such as DyGFormer, PINT, NAT, and CAWN, within a unique framework. Subsequently, they propose a novel model, TPNet, and conduct comparative evaluations using several datasets and competitors.

**Strengths:**

The provided framework offers a clearer and broader perspective on existing approaches based on temporal walks. The source code is well-commented and likely to be highly usable. The experimental setup investigates various competitors, datasets, and negative sampling techniques.

**Weaknesses:**

A fundamental concept in the article is "relative encoding." However, the intuition behind this concept is not introduced until line 119. I recommend presenting the meaning of "relative encoding" earlier in the article for better clarity and understanding.

**Questions:**

If I understand correctly, the authors introduce a weight decay mechanism to exponentially decrease the weight of older interactions. While this decision is well-motivated by previous studies, it is well known that the recurrence of interactions is fundamentally important in social networks (e.g., sociopatterns.org). I am curious whether the temporal periodicity of interactions is maintained with this approach. Could the authors discuss this aspect in more detail?

**Limitations:**

The authors discussed the limitations of the model in the appendix.

---

> ### Author Rebuttal · Authors · 2024-08-07
>
> Thanks for your insightful comments. In response, we have discussed the temporal periodicity modeling ability of our score function and moved the motivation of the relative encoding earlier according to your suggestion. We hope this addresses your concerns and are happy to answer any further questions you may have.
>
>
> **Q1. If I understand correctly, the authors introduce a weight decay mechanism to exponentially decrease the weight of older interactions. While this decision is well-motivated by previous studies, it is well known that the recurrence of interactions is fundamentally important in social networks (e.g., sociopatterns.org). I am curious whether the temporal periodicity of interactions is maintained with this approach. Could the authors discuss this aspect in more detail? .**
>
> Although the time decay score function (defined in line 187 of the paper) is not specifically designed for modeling temporal periodicity, it can still capture periodic patterns to some extent. To illustrate this, consider a simple example where two nodes, $u$ and $v$, interact once every $\Delta t$ time interval.  Then, we can divide the time into equal-size durations $(0,\Delta t),(\Delta t,2\Delta t),...$, and examine how the element of the temporal walk matrices changes within each duration.  Specifically, assuming the current time $t \in (n \Delta t, (n+1)\Delta t)$ with $n>0$, the timestamps of the historical interactions between $u$ and $v$ will be $[\Delta t, 2\Delta t,..., n\Delta t]$,  and the element of the one-step temporal walk matrix $A_{u,v}^{(1)}(t)$ will be $A_{u,v}^{(1)}(t) = \sum_{k=1}^n\exp(-\lambda (t-k\Delta t))$ (i.e., sum the scores of historical interactions). Since $\lambda >0$, $A_{u,v}^{(1)}(t)$ will decrease as $t$ increases within $(n\Delta t, (n+1)\Delta t)$ and the value of $A_{u,v}^{(1)}(t)$ is within $(\sum_{k=1}^n \exp(-\lambda k \Delta t),\sum_{k=0}^{n-1}\exp(-\lambda k \Delta t))$ (by setting $t$ to  $(n+1)\Delta t$ and $n\Delta t$ respectively). The maximum value $\sum_{k=0}^{n-1}\exp(-\lambda k \Delta t)$ is at least 1 since $\exp(-\lambda 0\Delta t) =1$ and all other terms are nonnegative. For the minimum value $\sum_{k=1}^n \exp(-\lambda k \Delta t)$,  if $\lambda$ is large enough, it will be less than 1. For example, if $\exp(-\lambda  \Delta t) < \frac{1}{4}$, we have $\sum_{k=1}^n \exp(-\lambda k \Delta t) < \sum_{k=1}^n \frac{1}{4^k} < \frac{1}{2}$. Based on the above analysis, $A_{u,v}^{(1)}(t)$ exhibits the following behavior over time: going up to a value that is no less than 1 at the beginning of each duration and then decreasing to a value that is lower than 1 at the end of each duration, which acts like a periodic function.  Then we can infer that $u$ and $v$ are likely to interact when $A_{u,v}^{(1)}(t)$ is low and unlikely to interact when $A_{u,v}^{(1)}(t)$ is high, capturing this periodic evolution pattern.  We show a visual illustration in Figure 1 of the attached PDF file for better understanding.
>
>
>
> **W1. A fundamental concept in the article is "relative encoding." However, the intuition behind this concept is not introduced until line 119. I recommend presenting the meaning of "relative encoding" earlier in the article for better clarity and understanding.**
>
> We have added the motivation for relative encoding to the second section of the Introduction and improved the explanation of Figure 1 for better clarity and understanding. The revised part of the second section is as follows (changes highlighted in bold):
>
> >  Relative encodings have become an indispensable module for effective temporal link prediction [6-9] where, without them, node representations computed independently by neighbor aggregation will fail to capture the pairwise information. As the toy example shown in Figure 1, A and F will have the same node representation due to sharing the same local structure. Thus it can not be determined whether D will interact with A or F at $t_3$ according to their representations. **However, by assigning nodes relative encodings (i.e., additional node features, detailed in Section 2.2) specific to the target link before computing the node representation, we can highlight the importance of each node and guide the representation learning process to extract pairwise information specific to the predicted link from the subgraph**. For example, in Figure 1, we can infer from the relative encoding of E **(in red circle)** that D is more likely to interact with F than with A since D and F share a common neighbor, E.

---

> > ### Comment · Reviewer_vApy · 2024-08-08
> >
> > Thank you for the detailed response. I’ve read it and you’ve clarified my doubt, so I’ve increased my score.

---

> > > ### Author Response · Authors · 2024-08-08
> > > **Response to Reviewer vApy**
> > >
> > > Thank you for your timely reply! We sincerely appreciate your support for our work.

---

### Author Rebuttal · Authors · 2024-08-07

We sincerely thank all reviewers for their time and valuable comments. We are pleased that the reviewers acknowledge the value of our work in providing a cohesive perspective on existing methods (Reviewers vApy, SMMz, YNEN), proposing a novel method (Reviewer vApy) with solid theoretical support (Reviewer SMMz), and conducting extensive experiments to verify the effectiveness and efficiency of the proposed method (Reviewers vApy, YNEN).

To the best of our efforts, we have provided thorough responses to address the issues raised by each reviewer, which mainly consist of:

- Clarification on method design and analysis of existing methods
  - Discussion on the temporal periodicity modeling capability of the score function.
  - Discussion on the construction of the function $g(\cdot)$.
  - Clarification of why ReLU can reduce estimation error.
  - Clarification on the use of temporal information in constructing temporal walk matrices.
- Additional experimental results
  - Reporting memory usage of different methods.
  - Comparison with a new temporal link prediction method.
- Reorganization of the paper.
  - Move the motivation of the relative encoding earlier.
  - Move the limitation section to the main body of the paper.

---

### Decision · Program_Chairs · 2024-09-25

**Decision:**

Accept (poster)

**Comment:**

The study explores the use of relative encodings for temporal link prediction. It first provides a unified view of relative encoding as a function of temporal walk matrices. Then it incorporates a time decay effect on this matrix, named TPNet. Finally, authors provide both theoretical and empirical analysis to demonstrate the effectiveness of the proposed approach.

All the reviewers are quite positive about this submission. Concerns raised by reviewers were addressed properly.

To sum up, it is a solid paper. I’d like to recommend its acceptance.